# Active information maintenance in working memory by a sensory cortex

Xiaoxing Zhang[1†], Wenjun Yan[1,2†], Wenliang Wang[1], Hongmei Fan[1], Ruiqing Hou[1], Yulei Chen[1], Zhaoqin Chen[1], Chaofan Ge[1,2], Shumin Duan[3,4], Albert Compte[5], Chengyu T Li[1]*

[1]State Key Laboratory of Neuroscience, CAS Center for Excellence in Brain Science and Intelligence Technology, Institute of Neuroscience, Shanghai Center for Brain Science and Brain-Inspired Technology, Chinese Academy of Sciences, Shanghai, China; [2]School of Future Technology, University of Chinese Academy of Sciences, Beijing, China; [3]Key Laboratory of Medical Neurobiology of Ministry of Health of China, Department of Neurobiology, Zhejiang University School of Medicine, Hangzhou, China; [4]Key Laboratory of Neurobiology of Zhejiang Province, Department of Neurobiology, Zhejiang University School of Medicine, Hangzhou, China; [5]Institut d'Investigacions Biomèdiques August Pi i Sunyer (IDIBAPS), Barcelona, Spain

**Abstract** Working memory is a critical brain function for maintaining and manipulating information over delay periods of seconds. It is debated whether delay-period neural activity in sensory regions is important for the active maintenance of information during the delay period. Here, we tackle this question by examining the anterior piriform cortex (APC), an olfactory sensory cortex, in head-fixed mice performing several olfactory working memory tasks. Active information maintenance is necessary in these tasks, especially in a dual-task paradigm in which mice are required to perform another distracting task while actively maintaining information during the delay period. Optogenetic suppression of neuronal activity in APC during the delay period impaired performance in all the tasks. Furthermore, electrophysiological recordings revealed that APC neuronal populations encoded odor information in the delay period even with an intervening distracting task. Thus, delay activity in APC is important for active information maintenance in olfactory working memory.
DOI: https://doi.org/10.7554/eLife.43191.001

*For correspondence:
tonylicy@ion.ac.cn

†These authors contributed equally to this work

Competing interests: The authors declare that no competing interests exist.

## Introduction

Working memory (WM) is a function of the brain that supports the active maintenance and manipulation of information over a delay period of several seconds (*Baddeley, 2012*). A buffer between recent external inputs and immediate behavioral outputs, WM is a critical component of cognition (*Jonides et al., 2008*). Originally the prefrontal cortex was thought to be the specific site for WM (*Jacobsen, 1935*). However, later results suggested that parallel distributed circuits are responsible for WM (*D'Esposito and Postle, 2015*; *Eriksson et al., 2015*; *Christophel et al., 2017*). Previously, we have shown that the delay-period activity of the medial prefrontal cortex (mPFC) of mice is only important during the learning but not the well-trained phase in an olfactory WM task (*Liu et al., 2014*). Here, we seek to elucidate the contribution of a sensory cortex to WM after mice are well-trained.

The role of sensory cortices in working memory is debated. On the one hand, the 'essential theory' stressed the importance of sensory cortices in WM (*Pasternak and Greenlee, 2005*;

*Scimeca et al., 2018*; *Gayet et al., 2018*), arguing that it is more parsimonious to receive, process, and maintain information in the same regions. Some recording and functional imaging experiments supported the hypothesis that sensory cortices can exhibit information-selective delay-period activity in WM tasks (*Pasternak and Greenlee, 2005*; *Fuster and Jervey, 1981*; *Miyashita and Chang, 1988*; *Harrison and Tong, 2009*; *Mendoza-Halliday et al., 2014*).Furthermore, perturbation of neural activity in sensory areas can impair WM performance (*Pasternak and Greenlee, 2005*; *Colombo et al., 1990*; *Harris et al., 2002*; *Seidemann et al., 1998*; *Guo et al., 2014*). On the other hand, the 'unessential theory' stressed that sustained activity emerges as a property of association cortices, but is not present in early sensory cortices, at least in the visual, somatosensory, and auditory domains (*Mendoza-Halliday et al., 2014*; *van Kerkoerle et al., 2017*; *Leavitt et al., 2017*). More importantly, a distractor presented during the delay period diminished delay-period sustained activity in sensory cortices without changing performance (*Miller et al., 1996*; *Bettencourt and Xu, 2016*), arguing that sensory cortices may not be important for WM tasks (*Xu, 2018*).

The two theories make opposite predictions in perturbation and recording experiments. If a sensory cortex is important, behavioral performance should be impaired following suppression of delay-period activity in a sensory region in a temporally precise manner. Moreover, in a task with distractors during the delay, the neuronal activity of sensory regions should be able to maintain information even following a distractor. If a sensory cortex is not important, both experiments should produce negative results. In the current study, we specifically tested these predictions in the olfactory domain in behaving mice, using optogenetic perturbation (*Fenno et al., 2011*) and electrophysiological recordings with the necessary temporal specificity.

To engage active WM maintenance, we trained mice to perform several WM tasks, including a dual-task paradigm. In this task, we inserted a distracting task into the delay period of an ongoing WM task. To perform the dual task successfully, mice need to maintain the sample information of the outer WM task while performing the distracting task. Such a dual-task design explicitly challenges the central executive control ability of a subject (reviewed by *Baddeley, 2012* and *Watanabe and Funahashi, 2018*). A hallmark of the dual task is that performance in either of the component tasks is reduced relative to that trained in isolation, termed dual-task interference. Previously, the neural correlates of dual-task interference have been reported in the neuronal activity of monkey prefrontal cortex (*Watanabe and Funahashi, 2014*) and in results from human functional imaging (*D'Esposito et al., 1995*). Patients with a frontal-cortex lesion are also impaired in this type of task (*Baddeley et al., 1997*), but the functional role and neural correlates of sensory cortices in dual-task performance remain unknown.

In the current study, we focused on the anterior piriform cortex (APC) for olfactory WM. The APC is a good candidate because it is directly connected with the olfactory bulb (*Ghosh et al., 2011*; *Miyamichi et al., 2011*; *Sosulski et al., 2011*; *Bekkers and Suzuki, 2013*; *Price and Powell, 1970*; *Davison and Ehlers, 2011*), encodes odor information (*Bekkers and Suzuki, 2013*; *Schoenbaum and Eichenbaum, 1995*; *Wilson, 1998*; *Poo and Isaacson, 2009*; *Stettler and Axel, 2009*; *Shusterman et al., 2011*; *Miura et al., 2012*; *Courtiol and Wilson, 2017*; *Zhan and Luo, 2010*; *Gire et al., 2013*; *Iurilli and Datta, 2017*; *Bolding and Franks, 2017*; *Otazu et al., 2015*), and is important for olfactory behavior (*Bekkers and Suzuki, 2013*; *Miura et al., 2012*; *Courtiol and Wilson, 2017*; *Choi et al., 2011*). The rich recurrent connectivity within the APC (*Bekkers and Suzuki, 2013*; *Franks et al., 2011*; *Bolding and Franks, 2018*) is also well suited for generating reverberating activity that outlasts sensory inputs (*Major and Tank, 2004*). In addition, activity patterns of the APC neurons are critically dependent on olfactory experience, indicating an active involvement (*Wilson, 1998*; *Courtiol and Wilson, 2017*; *Saar et al., 1999*; *Roesch et al., 2007*). Finally, the piriform cortex is also activated in a human olfactory WM task (*Zelano et al., 2009*). Here, we addressed the role of the APC in olfactory WM using a combination of optogenetics and electrophysiology methods. We found that the APC plays a prominent role in this task and that its neural activity reflects the maintenance of task parameters through the WM delay.

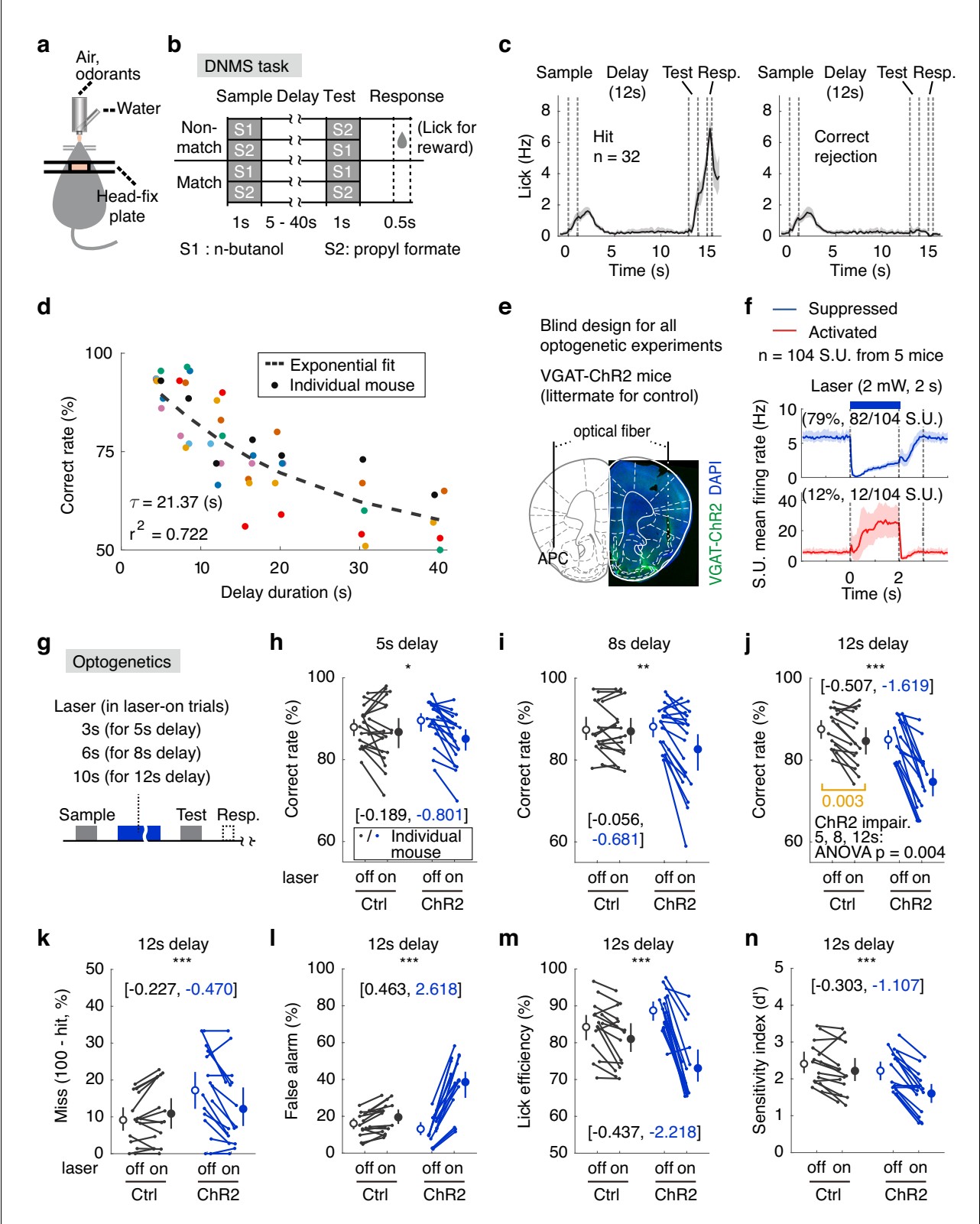

**Figure 1.** The APC activity is important for the DNMS performance. (a) Diagram of the experimental setup. (b) Design of the DNMS task. (c) Averaged licking rate of mice well-trained for the DNMS task in hit (left) and correct rejection (right) trials. Shadow, 95% confidence interval of the mean from bootstrapping of 1000 repeats. (d) The DNMS task correct rate with varying delay durations, fitted by an exponential-decay function. Each color represents the same individual mouse across all delay periods. (e) Expression of ChR2, targeted region and sites of optical fiber insertion. (f) The

*Figure 1 continued on next page*

*Figure 1 continued*

majority of neurons (single units) were silenced during optogenetic suppression, in awake single unit op-tetrode recording in vivo. Numbers above figure are the numbers of single units. The apparent slower increase in firing for the activated neurons is due to the large variation in firing modulation, also see *Figure 1—figure supplements 3* and *4*. (g) Diagram of the optogenetic perturbation during the delay period. (h–j) Correct rates in the DNMS task following delay-period optogenetic suppression in APC. Each point in the figure represents the averaged result from one mouse. *, p=0.023, **, p=0.002, ***, p<0.001, mixed-between-within-ANOVA, genotype × laser interaction; error bars show the 95% confidence intervals of the means from bootstrapping of 1000 repeats. Numbers in brackets show the effect size measured by Cohen's d for control (in black) and ChR2 (in blue) groups. Delay duration, 5 s (h), 8 s (i) and 12 s (j). In the 12s-delay DNMS task, the performance in control mice were also modestly impaired by laser stimulation ((j) in orange, paired permutation test). (k–n) Miss rate, false alarm rate, lick efficiency and sensitivity index (d') following APC delay-period suppression in the DNMS task with 12 s delay. ***, p<0.001, mixed-between-within-ANOVA, genotype × laser interaction; error bars show the 95% confidence interval of the means from bootstrapping of 1000 repeats. Numbers in brackets show the effect size measured by Cohen's d for control (in black) and ChR2 (in blue) groups. See *Figure 1—figure supplements 1–6*, *Figure 1—source data 1*, *2* for more details of the task and complete statistics.

DOI: https://doi.org/10.7554/eLife.43191.002

The following source data and figure supplements are available for figure 1:

**Source data 1.** Performance in the DNMS task.
DOI: https://doi.org/10.7554/eLife.43191.009
**Source data 2.** Effect size and ANOVA statistics for optogenetics.
DOI: https://doi.org/10.7554/eLife.43191.010
**Figure supplement 1.** Odor residual of the odor-delivery system.
DOI: https://doi.org/10.7554/eLife.43191.003
**Figure supplement 2.** Location for optical fibers and range of optogenetic stimulation.
DOI: https://doi.org/10.7554/eLife.43191.004
**Figure supplement 3.** Single-unit spike-sorting quality.
DOI: https://doi.org/10.7554/eLife.43191.005
**Figure supplement 4.** Optogenetic effects on APC neurons.
DOI: https://doi.org/10.7554/eLife.43191.006
**Figure supplement 5.** The APC activity is important for the DNMS performance.
DOI: https://doi.org/10.7554/eLife.43191.007
**Figure supplement 6.** Lick rate in the DNMS tasks.
DOI: https://doi.org/10.7554/eLife.43191.008

# Results

## The design and performance of a delayed non-match to sample task

To temporally dissociate information maintenance from perception and decision making, we first trained head-fixed mice to perform an olfactory delayed non-match to sample (DNMS) task (*Liu et al., 2014*), using an automatic training system (*Han et al., 2018*). In this task, two odors (S1, n-butanol; or S2, propyl formate) were presented to mice, separated by a delay period (5–40 s, *Figure 1a,b*, *Video 1*). The first odor residue was briskly cleared (*Figure 1—figure supplement 1*, the delay-period residual concentration is lower than the sensory threshold for this task [*Liu et al., 2014*]) at the beginning of the delay period. Licking following the non-match trials (S1-S2, or S2-S1) led to water reward. No explicit punishment was applied for the error trials. Mice readily performed the task with little licking during the delay period (*Figure 1c*, *Video 1*, *Video 2*). A hallmark of WM is a progressive decay of performance with increasing delay duration (*Baddeley, 2012*), which was indeed observed in the task (*Figure 1d*).

## APC delay-period activity is important for olfactory DNMS task performance

It has previously been shown that the delay-period activity of the medial prefrontal cortex (mPFC) is important only during learning of the DNMS task, but not when mice are well-trained (*Liu et al., 2014*). To test the potential role of the APC in the well-trained phase, we optogenetically (*Fenno et al., 2011*) suppressed the delay-period activity of APC pyramidal neurons after mice were well-trained (defined as correct rate above 80% in 40 consecutive trials). Transgenic mice expressing channel-rhodopsin (ChR2) only in GABAergic neurons [with the promoter of the *Slc32a1* gene that encodes vesicular GABA transporter (VGAT), referred to as VGAT-ChR2 (*Zhao et al., 2011*)] were used as the experimental group, whereas the ChR2-negative littermates were used as the control

group in a blind design (*Figure 1e*, see Materials and methods). The VGAT-ChR2 mice have been used extensively in many studies to suppress local excitatory neurons (e.g. *Liu et al., 2014*; *Guo et al., 2014*). The expression and effectiveness of ChR2 were verified by immunostaining and by op-tetrode single-unit recording, respectively (*Figure 1e* and *Figure 1—figure supplement 2a and b*). The specificity of optogenetic suppression was verified by c-*Fos* labeling (*Figure 1—figure supplement 2c*). The majority of the APC neurons (79%, 82 out of 104) were suppressed during the optogenetic stimulation, while a small fraction (12%, 12 out of 104) were activated; the optogenetic effect diminished within 1 s after laser-offset (see the third dotted line marked 1 s after laser-offset in *Figure 1f*; spike-sorting quality in *Figure 1—figure supplement 3*; and example single neurons showing activity modulation induced by optogenetic perturbation in *Figure 1—figure supplement 4*). Activity suppression was achieved by step-laser illumination (473 nm, 2 mW) during laser-on trials, each followed by a laser-off trial (*Figure 1g*). Performance was impaired following optogenetic suppression of the APC delay-period activity in the VGAT-ChR2 mice, with delay duration of 5, 8, or 12 s (*Figure 1h–j*; for effect size and complete ANOVA statistics, see *Figure 1—figure supplement 5i–m* and *Figure 1—source data 1*, *2*). Moreover, optogenetic suppression in trials with longer delay duration resulted in larger deficits (*Figure 1h–j*).Note that the performance of control mice could also be affected by light illumination with longer duration, for example 10 s laser-on condition in the DNMS task with 12 s delay, probably through distraction by visual illumination. However, this was controlled by the ANOVA interaction statistics. Behavioral deficits were reflected as an increased false-alarm rate and a decreased discriminability (*d'*) (false alarm rate for all delay durations, *d'* for 8 s and 12 s delay durations, *Figure 1l,n*, *Figure 1—figure supplement 5, b,f and h*, *Figure 1—source data 1*, *2*). Similarly, the lick efficiency (the percentage of successful licks with reward) was impaired in laser-on trials in the VGAT-ChR2 mice (*Figure 1m*, *Figure 1—figure supplement 5, c and g*, *Figure 1—source data 1*, *2*). The ability of the mice to lick during the response window, as well as outside the response window, was otherwise not affected by laser illumination (*Figure 1—figure supplement 6, a–e*). Note that the mice exhibited transient anticipatory licking only in the trials with fixed association between laser-offset and test-onset. The differences are small and transient, and are unlikely to affect the quantification of behavioral response during the response window.

## Statistical interaction between duration of delay period and timing of optogenetic suppression

To further clarify the interactions among delay duration, laser duration, and laser onset/offset, we performed two more sets of optogenetic experiments. In the first set of experiments, the delay duration of the DNMS task was fixed at 12 or 20 s but the laser onset and duration were systematically changed (*Figure 2a,b–c* for 12 s, *Figure 2d,e* for 20 s delay duration). Specifically, optogenetic suppression of APC neuronal activity was varied in both duration (3, 6, 10, or 12 s) and temporal position (early, mid or late in the delay period) in a trial-by-trial fashion. A visual mask was applied to diminish the visual distraction induced by laser illumination (as seen in *Figure 1j*). Optogenetic suppression significantly impaired the DNMS task performance in VGAT-ChR2 mice, especially for laser

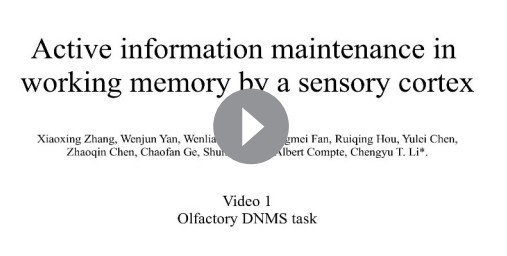

**Video 1.** Performance of a delayed non-match to sample task in mice.
DOI: https://doi.org/10.7554/eLife.43191.011

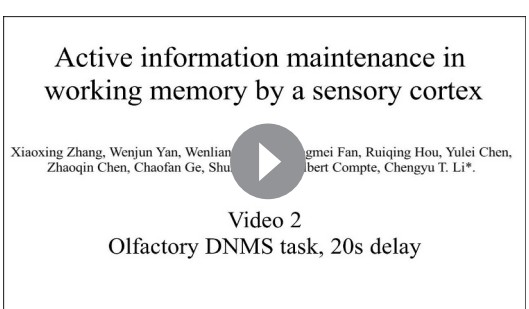

**Video 2.** Continuous video recording of mice performing a DNMS task with 20s-delay.
DOI: https://doi.org/10.7554/eLife.43191.012

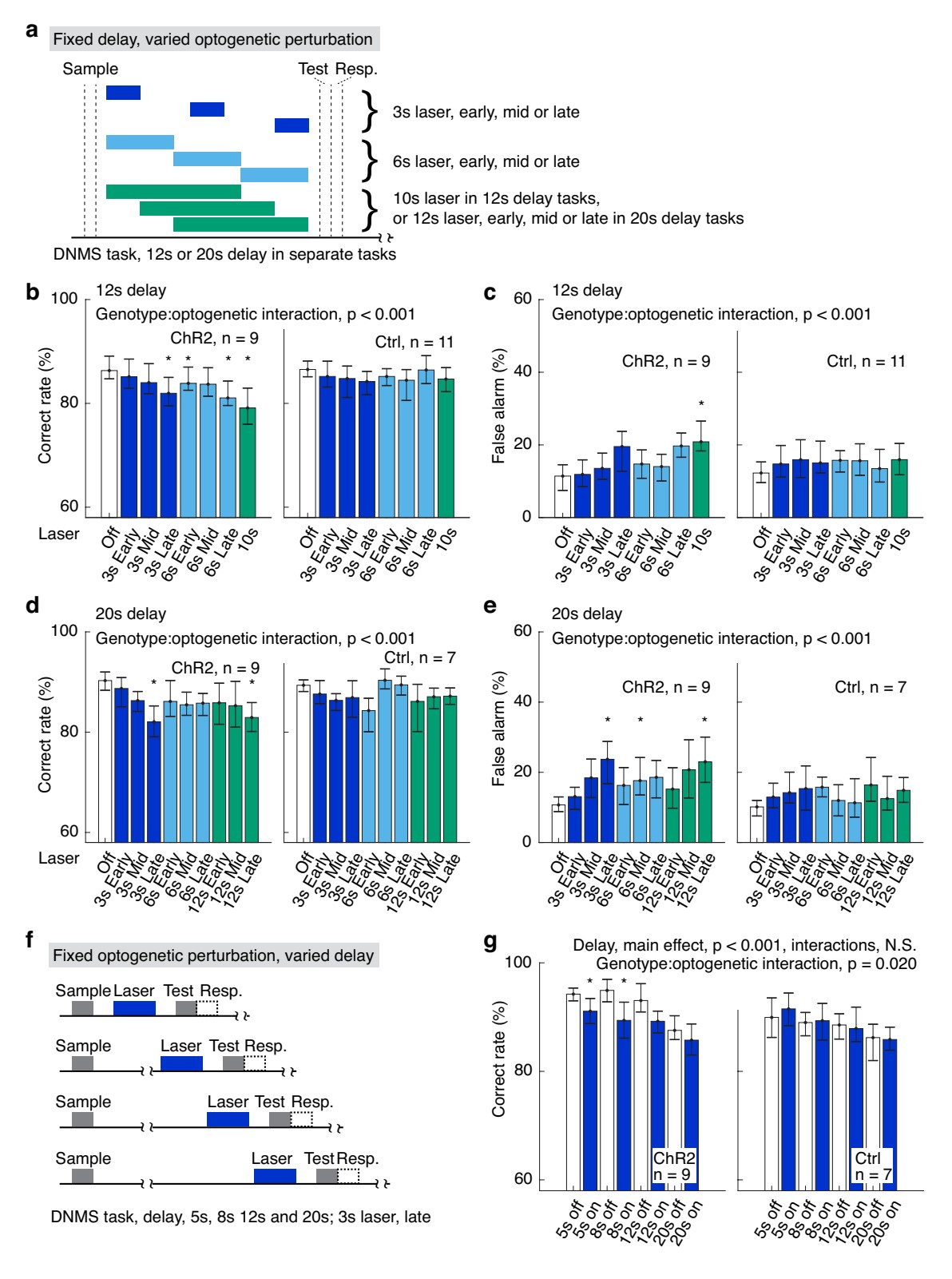

**Figure 2.** Varied delay and optogenetic suppression effects. (**a**) Designs of the fixed delay, varied optogenetic suppression task. (**b, c**) Correct rates (**b**) and false alarm rates (**c**) in the fixed delay, varied optogenetic suppression task with 12 s delay. (**d, e**) As in (**b**) and (**c**), but for tasks with 20 s delay duration. (**f**) Design of the fixed optogenetic suppression, varied delay task. (**g**) Correct rates in the fixed optogenetic suppression, varied delay task; interactions for delay include delay:genotype, delay:laser-on/off, and delay:genotype:laser-on/off, all of which are statistically not significant (N.S.).
*Figure 2 continued on next page*

*Figure 2 continued*

Statistics at the top of each panel were generated by mixed-between-within-ANOVA. *, p<0.05 in a post-hoc paired permutation test of 10,000 repeats comparing to laser-off control, adjusted for multiple comparison with the Bonferroni method. For complete statistics, see *Figure 2—source data 1,3*. For more details of the optogenetic task performance and effect size, see *Figure 2—figure supplements 1–2*.

DOI: https://doi.org/10.7554/eLife.43191.013

The following source data and figure supplements are available for figure 2:

**Source data 1.** Fixed delay, varied optogenetic suppression.

DOI: https://doi.org/10.7554/eLife.43191.016

**Source data 2.** Fixed optogenetic suppression, varied delay duration.

DOI: https://doi.org/10.7554/eLife.43191.017

**Source data 3.** Effect size and ANOVA statistics for optogenetics.

DOI: https://doi.org/10.7554/eLife.43191.018

**Figure supplement 1.** Varied delay and optogenetic suppression effects.

DOI: https://doi.org/10.7554/eLife.43191.014

**Figure supplement 2.** Optogenetic suppression effect size.

DOI: https://doi.org/10.7554/eLife.43191.015

---

illumination during the late-delay period (*Figure 2b,d*, *Figure 2—source data 1,2*). False alarm rate was also significantly increased in the VGAT-ChR2 mice in similar conditions (*Figure 2c,e*). Miss rate was not affected by the optogenetic suppression in both VGAT-ChR2 and control mice (*Figure 2—figure supplement 1a and b*, *Figure 2—source data 1,2*). Furthermore, in the laser-on trials for VGAT-ChR2 mice, the duration and timing of laser illumination had a significant interaction in determining the task correct rates, false alarm rates and miss rates (*Figure 2b–e*, *Figure 2—source data 1,2*). The effect size of the above results is shown in *Figure 2—figure supplement 2*. Therefore, optogenetic suppression of longer laser-on duration during a later phase of the delay period results in larger behavioral defects.

In the second set of experiments, the optogenetic suppression was kept at 3 s in duration and within the late delay period, while the delay duration was varied between 5 s and 20 s, in a trial-by-trial fashion (*Figure 2f*). We observed significant effect of delay duration on DNMS task performance, independent of genotype (*Figure 2g*, *Figure 2—figure supplement 1c and d*, *Figure 2—source data 2,3*). Consistent with previous results, optogenetic suppression impaired task performance only in VGAT-ChR2 mice (*Figure 2g*, *Figure 2—figure supplement 1c and d*, *Figure 2—source data 2,3*). We did not observe significant interaction between delay-period duration and laser on/off (*Figure 2g*, *Figure 2—figure supplement 1c and d*, *Figure 2—source data 2,3*). Therefore, longer delay duration does not result in different behavioral impairment induced by optogenetic suppression of fixed duration.

## Performance impairment induced by delay-period optogenetic suppression is not due to impaired perception during the test-odor delivery period

The optogenetic manipulation during the delay period might impair sensory perception during the test-odor delivery period. To exclude this possibility, we suppressed APC activity before odor delivery in three tasks. The logic of the experiments was to recruit perception in similar tasks that do not require delay-period WM. If no behavioral deficit was observed by silencing APC activity, any impaired performance in the DNMS task should not be due to processes unrelated to delay-period WM maintenance. In all three tasks, we observed intact behavioral performance (*Figure 3a–f*, *Figure 3—source data 1–2*; effect size, *Figure 3—figure supplement 1b*). First, we optogenetically suppressed APC activity before the sample-delivery period (baseline) for 3 s (delay duration 5 s), 6 s and 10 s (delay duration 12 s) in the DNMS task, matching the delay-period optogenetic suppression (*Figure 3a–c*). Second, mice were trained to perform a sensory-discrimination Go/No-go (GNG) task (*Figure 3d*). The APC activity before the test odor was optogenetically suppressed (*Figure 3d and e*). Third, mice were trained to perform a non-match-to-sample without delay (NMS-WOD) task (*Figure 3d*). The essence of the design of this task is that the decision is based on matching the relationship between sample and test odors, as in the DNMS task. The third task takes longer to train (~300 trials) than the Go/No-go task, which can be learned within 100 trials in one day

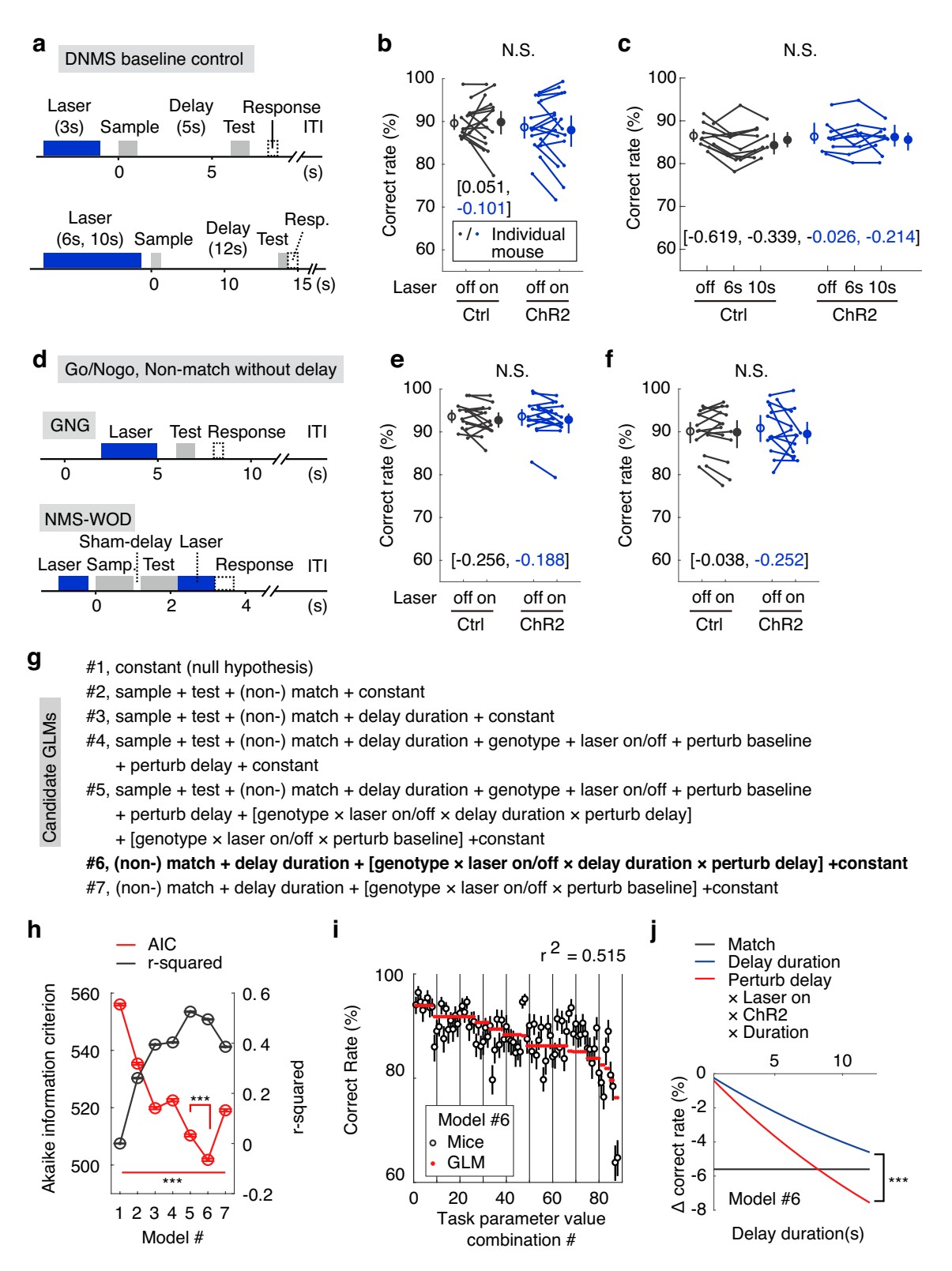

**Figure 3.** The optogenetic suppression effects were not due to impaired sensory perception. (a) Designs of the DNMS-baseline optogenetic control task. (b, c) Correct rates in the DNMS-baseline perturbation control task with 3 s perturbation and 5 s delay (b) and with 6 s and 10 s perturbation and 12 s delay (c). N.S., statistically not significant, mixed-between-within-ANOVA, genotype × laser interaction; error bars represent the 95% confidence interval of the mean from bootstrapping of 1000 repeats. Numbers in brackets are the effect size measured by Cohen's d for control (in black) and

*Figure 3 continued on next page*

*Figure 3 continued*

ChR2 (in blue) groups. (d) Designs of the Go/No-go (GNG) and non-match to sample without delay (NMS-WOD) optogenetic control tasks. (e, f) Correct rates in the GNG and NMS-WOD control tasks. N.S., statistically not significant, mixed-between-within-ANOVA, genotype × laser interaction; error bars represent the 95% confidence interval of the mean from bootstrapping of 1000 repeats. Numbers in brackets are the effect size measured by Cohen's d for control (in black) and ChR2 (in blue) groups. (g) Design of the candidate generalized linear models (GLMs). The task parameters in each model were then used to fit task performance. Terms in square brackets represent interactions. (h) Comparison of the candidate GLMs in terms of Akaike information criterion (AIC) and r-squaredin explaining performance, between all models, and between model 5 and 6. ***, p<0.001, in a permutation test of 1000 repeats. (i) Performance of mice in the WM tasks versus the optimized model (#6 in (g)) coefficients were calculated from all experimental trials, see Materials and methods). Error bars represent 95% confidence intervals of means from bootstrapping of 1000 repeats. (j) Comparison showing the predicted effect size against delay duration, with coefficients derived from parameters in model #6. Shadow represents the 95% confidence intervals of means from bootstrapping of 1000 repeats (the confidence interval is very narrow and may appear invisible). ***, p<0.001 in comparisons of the coefficients of delay-duration and the effect of delay-period laser perturbation (interaction among perturb-delay, laser-on, ChR2-genotype and delay-duration, see Materials and methods) in the model, permutation test with 1000 repeats. See *Figure 3—figure supplement 1*, for the comparison of the learning curve in GNG and NMS-WOD tasks and comparison of the effect size of optogenetic suppression. See *Figure 3—source data 1–4* for complete statistics.

DOI: https://doi.org/10.7554/eLife.43191.019

The following source data and figure supplement are available for figure 3:

**Source data 1.** The optogenetic suppression effects were not due to impaired sensory perception.
DOI: https://doi.org/10.7554/eLife.43191.021
**Source data 2.** Effect size and ANOVA statistics for optogenetics.
DOI: https://doi.org/10.7554/eLife.43191.022
**Source data 3.** General linear model coefficients.
DOI: https://doi.org/10.7554/eLife.43191.023
**Source data 4.** Task parameter combinations and general linear model fit.
DOI: https://doi.org/10.7554/eLife.43191.024
**Figure supplement 1.** Learning curve for the GNG and NMS-WOD tasks.
DOI: https://doi.org/10.7554/eLife.43191.020

(*Figure 3—figure supplement 1*), suggesting a higher level of attention and effort. However, there is a minimal delay between odors, and therefore no requirement for information maintenance. APC activity before and after test-odor delivery was suppressed in the task (*Figure 3d and f*). The lack of behavioral impairment in all three control experiments excluded the perception-impairment hypothesis (*Figure 3—source data 1–2*). Therefore, the behavioral deficits following optogenetic perturbation of APC delay-period activity reflected the contribution of this activity to information maintenance in the DNMS task.

## Formal model comparison quantitatively demonstrating the importance of APC delay-period activity

In order to quantify the specific contribution of APC delay-period activity to performance, when compared to other explaining task parameters, we designed seven candidate generalized linear models (GLMs) by systematically varying the combinations of task parameters in fitting the DNMS and control task performance (*Figure 3g*). A good model should exhibit a high coefficient of determination in explaining performance (r-squared) and a low Akaike information criterion (AIC, see Materials and methods), which punishes a higher number of free parameters. Such formal model comparison can quantitatively reveal the relative contributions of different behavioral parameters in determining performance (*Hwang et al., 2017*; *Brunton et al., 2013*).

We started with the null-hypothesis that no task parameter affects performance (#1, *Figure 3g,h*). Adding the variables of both the sensory cues (#2) and the time constant for memory decay (#3, from *Figure 1d*) improved the performance of the models, consistent with the obvious importance of sensory cues and delay duration in this task. The genotypes, laser on/off, and the perturbation in delay/baseline period did not further improve the model if added individually (#4). However, the interaction among all these terms improved the model (#5), consistent with impaired performance by optogenetic suppression during the delay-period for the VGAT-ChR2 mice, but not during the baseline period or for the control mice. By eliminating less-predictive variables, we obtained the optimal model (#6, AIC value significantly smaller than all other models; *Figure 3g–i*, *Figure 3—source data 3 and 4*), which contained only four key parameters and the interactions among them:

optogenetic suppression or not during the delay period, match or non-match relationship, and the delay duration (coupled with laser-on duration). This association between the WM delay perturbation and the delay duration is consistent with the important role of APC delay activity in information maintenance (*Figure 3j*). Quantitatively, the impact of optogenetic perturbation out-weighted that of delay duration (*Figure 3j*). The above analysis further demonstrated the importance of APC delay-period activity for WM in the olfactory DNMS task.

## The importance of the APC delay-period activity in an olfactory delayed paired association task

The specificity of the DNMS task in testing WM maintenance can be disputed, especially in relation to sensory cortices. For example, it has also been suggested that the DNMS task in monkeys can be performed through a recency effect and with little selective memory maintenance (*Wittig and Richmond, 2014*). Furthermore, the activity of a sensory cortex can be suppressed or enhanced following the repetition of a sensory stimulus (*Miller et al., 1993*). Therefore, one can argue that sensory adaptation, but not WM, may be perturbed by optogenetic manipulation in this task. To eliminate the involvement of repeating sample cues, we trained mice to perform an olfactory delayed paired association (DPA) task (*Figure 4a*). In each trial of this task (modified from *Schoenbaum and Eichenbaum, 1995*), a sample odor (S1, ethyl acetate; or S2, 3-methyl-2-buten-1-ol) was presented, followed by a delay period (13 s), then a test odor (T1, n-butanol; or T2, propyl formate). Mice were rewarded with water if they licked within the response window in the paired trials (S1-T1 or S2-T2), but not in the unpaired trials (S1-T2 or S2-T1, *Figure 4a*). Successful performance of the task required WM maintenance and learning of the arbitrary association between the odor pairs. Importantly, repetition (*Miller et al., 1993*) or recency effects (*Wittig and Richmond, 2014*) cannot be used to solve the task, because the sample odor is not repeated within a trial. Mice readily learned the task (*Figure 4d*, black curve). Critically, optogenetic suppression of APC delay-period activity impaired DPA performance (*Figure 4b*, *Figure 4—figure supplement 1a–d*, *Figure 4—source data 1,2*). The effect size of the above results is shown in *Figure 4—figure supplement 2*. Although optogenetic impairment in DPA performance (difference in correct rate between laser-on and laser-off trials) varied across mice, we did not find outliers with two widely used statistical methods: (1) for elements more than 1.5 interquartile ranges above the upper quartile or below the lower quartile (*Hattori et al., 2017*; *Kato et al., 2013*; *Sreenivasan et al., 2016*); and (2) Grubb's method (*Namburi et al., 2015*; *Burgos-Robles et al., 2017*; *Carus-Cadavieco et al., 2017*).

Therefore, the APC delay-period activity is important for WM maintenance, even in a task without repeating sensory cues.

## APC delay activity is important for active maintenance against a distracting task

A hallmark of active maintenance in WM is resistance against distractors during the delay period (*Baddeley, 2012*; *Miller et al., 1996*; *Bettencourt and Xu, 2016*). To test this ability in mice, we added a distracting GNG task during the delay period of the DPA task (*Figure 4d*). This paradigm belongs to the dual-task designs that have been used to study central executive control (*Baddeley, 2012*; *Watanabe and Funahashi, 2018*; *Watanabe and Funahashi, 2014*), because mice are required to split attention in the middle of the delay period in order to perform the GNG task, while simultaneously maintaining the sample information of the DPA task (*Figure 4c*). The design of this task also ensured that any lingering residual sample odor would be flushed by the concentrated olfactory cues of the inner task, therefore the WM but not residual odor is required for the outer-task performance. Mice were trained to perform the DPA task, then the dual-task (*Figure 4d,e*; red and blue curves). After the initial drop in DPA performance, mice learned to perform dual-task well (*Figure 4d,e*). DPA performance in the dual-task paradigm was worse than that in the simple DPA task even in the well-trained phase (*Figure 4d,f*), consistent with the dual-task interference observed in humans (*Baddeley, 2012*) and monkeys (*Watanabe and Funahashi, 2014*). Moreover, interference was dependent on the trial types inserted in the DPA delay, with the worst performance for the Go-distractor trials (*Figure 4f*). We then optogenetically suppressed the delay-period activity of APC pyramidal neurons after the distracting GNG task. The DPA false-alarm rate within the dual task increased significantly in the laser-on trials in the VGAT-ChR2 group (*Figure 4h*). The overall

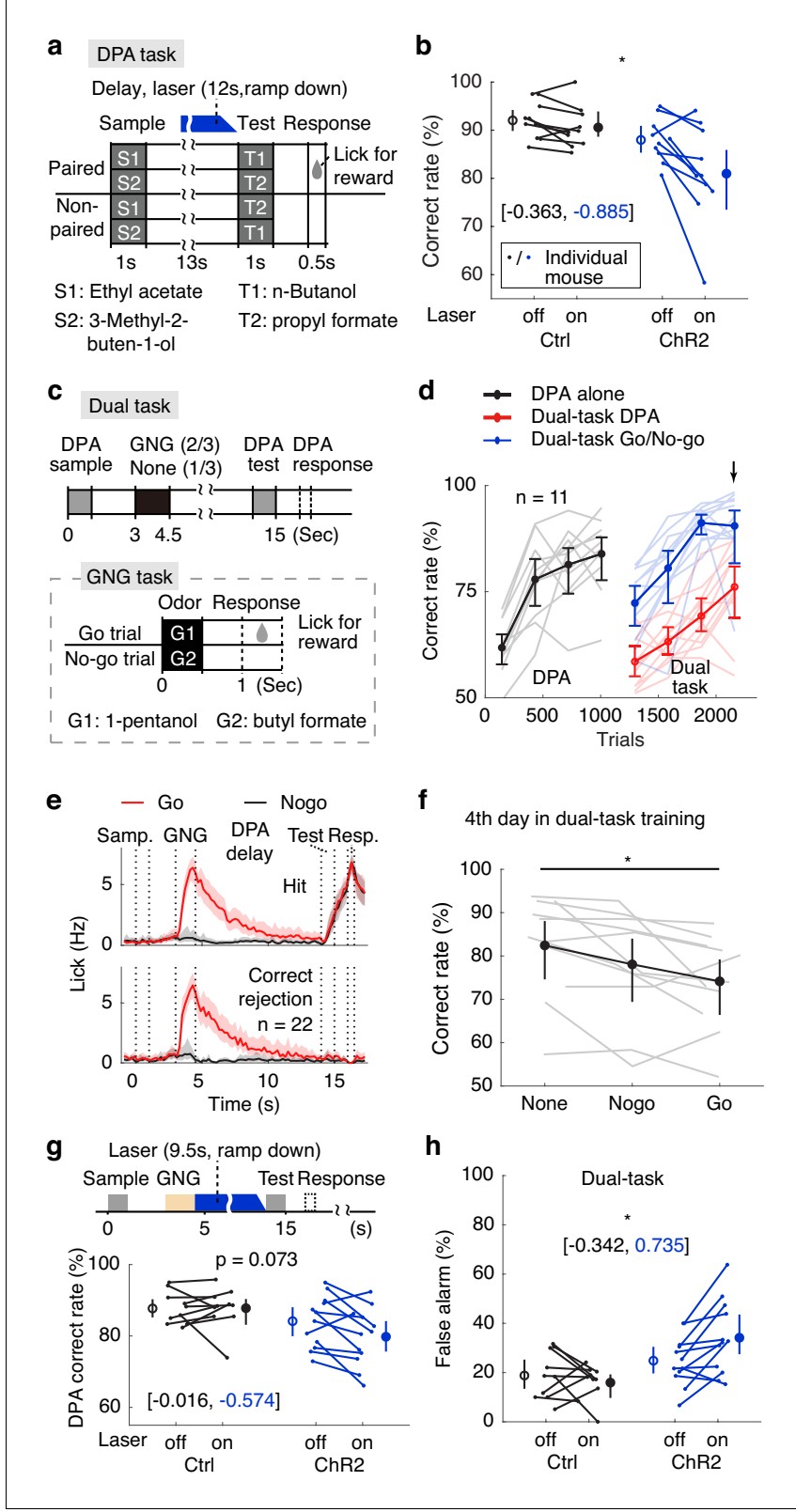

**Figure 4.** Active memory maintenance by the APC delay activity. (**a**) Design of the delayed paired association (DPA) task. (**b**) Correct rates following APC delay-period optogenetic suppression in the DPA task. *, p=0.038, as determined by a mixed-between-within-ANOVA, genotype × laser interaction; error bars represent the 95% confidence intervals of the means from bootstrapping of 1000 repeats. Numbers in brackets show the effect size

*Figure 4 continued on next page*

*Figure 4 continued*

measured by Cohen's d for control (in black) and ChR2 (in blue) groups. We noticed some performance variations resulting from DPA optogenetic impairment, and confirmed that there is no outlier with both the 1.5 interquartile ranges and the Grubb's methods. (c) Behavioral diagram for the dual-task design. (d) Learning curve for the correct rate in the dual-task. Note the drop of DPA performance after inserting the GNG task in the delay period (in red). Arrow, the 4th day in dual-task training. Light-color traces show the data for individual mice; error bars represent the 95% confidence intervals of the means from bootstrapping of 1000 repeats. (e) Averaged licking rate of mice well-trained for the dual-task in GNG-Go (red), GNG-Nogo (black), DPA-hit (top) and DPA-correct rejection (bottom) trials. Shadows show the 95% confidence interval of the mean from bootstrapping of 1000 repeats. (f) Dual-task interference as illustrated by performance data from the 4th day of dual-task training. *, p=0.041, one-way repeated measure ANOVA. Gray traces show data for individual mice; error bars show the 95% confidence interval of the mean from bootstrapping of 1000 repeats. (g) DPA-Correct rate after suppressing the APC activity during the later-phase delay period after distractors. P values were obtained from mixed-between-within-ANOVA, genotype × laser interaction; error bars show the 95% confidence intervals of the means from bootstrapping of 1000 repeats. Numbers in brackets represent the effect size as measured by Cohen's d for control (in black) and ChR2 (in blue) groups. (h) As in (g), but for DPA-false alarm rate. *, p=0.014. See *Figure 4—figure supplement 1* for more details for the task performance and control tasks. See *Figure 4—figure supplement 2* for optogenetic suppression effect size. See *Figure 4—source data 1–2* for complete statistics.
DOI: https://doi.org/10.7554/eLife.43191.025

The following source data and figure supplements are available for figure 4:

**Source data 1.** Active memory maintenance by the APC delay activity.
DOI: https://doi.org/10.7554/eLife.43191.028

**Source data 2.** Effect size and ANOVA statistics for optogenetics.
DOI: https://doi.org/10.7554/eLife.43191.029

**Figure supplement 1.** Active memory maintenance by the APC delay activity.
DOI: https://doi.org/10.7554/eLife.43191.026

**Figure supplement 2.** Optogenetic suppression effect size.
DOI: https://doi.org/10.7554/eLife.43191.027

---

performance also tended to differ (genotype-laser interaction p=0.07 in a mixed-between-within-ANOVA). As a negative control, optogenetic suppression before sample delivery did not affect performance (*Figure 4—figure supplement 1i–n*). The effect size of the above results is shown in *Figure 4—figure supplement 2*. Therefore, APC delay-period activity is important for active maintenance in the face of distraction from an intervening task.

## Neural correlates of APC activity in the DNMS task

To investigate the neural correlates of the APC in WM, we recorded single-unit activity using custom-made tetrodes while mice were performing WM tasks (tetrodes, *Figure 5—figure supplement 1a–c*; example neurons, *Figure 5a and b*; spike-sorting quality in *Figure 1—figure supplement 3*) (*Liu et al., 2014*). In the DNMS task, mice were trained with a 4 s delay period. Recording was started from the first day of the training. Recording electrodes were advanced daily (approximately 50 μm/day after recording sessions). After 2–4 days into the training, the delay duration was increased to 8 s. We recorded 204 neurons from 19 mice while they performed the task with 4 s delay duration and 156 neurons from 20 mice with 8 s delay duration. A subset of APC neurons showed selective activity following exposure to different sample odors during the delay period in correct trials (for examples, see *Figure 5a–c*, *Figure 5—figure supplement 2a–i*). Some of these neurons were able to code for the sample information in single trials (*Figure 5c and d*, *Figure 5—figure supplement 2c,f and i*, as measured by the area under receiver operating characteristic, auROC). The distribution of the auROC for all neurons is plotted in *Figure 5—figure supplement 2j and k*. In error trials, the coding ability of these neurons was reduced (as measured by auROC, *Figure 5c and d*, *Figure 5—figure supplement 2b-i*). Thus, APC neuronal activity encodes WM information and is correlated with task performance.

Significant coding for the maintained information in APC neuronal activity was further revealed in neuronal-activity heat maps, separately plotted for different sample odors (8 s and 4 s delay duration in *Figure 5e* and *Figure 5—figure supplement 3a*, respectively), or for the firing-rate selectivity index (*Figure 5f* and *Figure 5—figure supplement 3b*, see Materials and methods). More than 25%

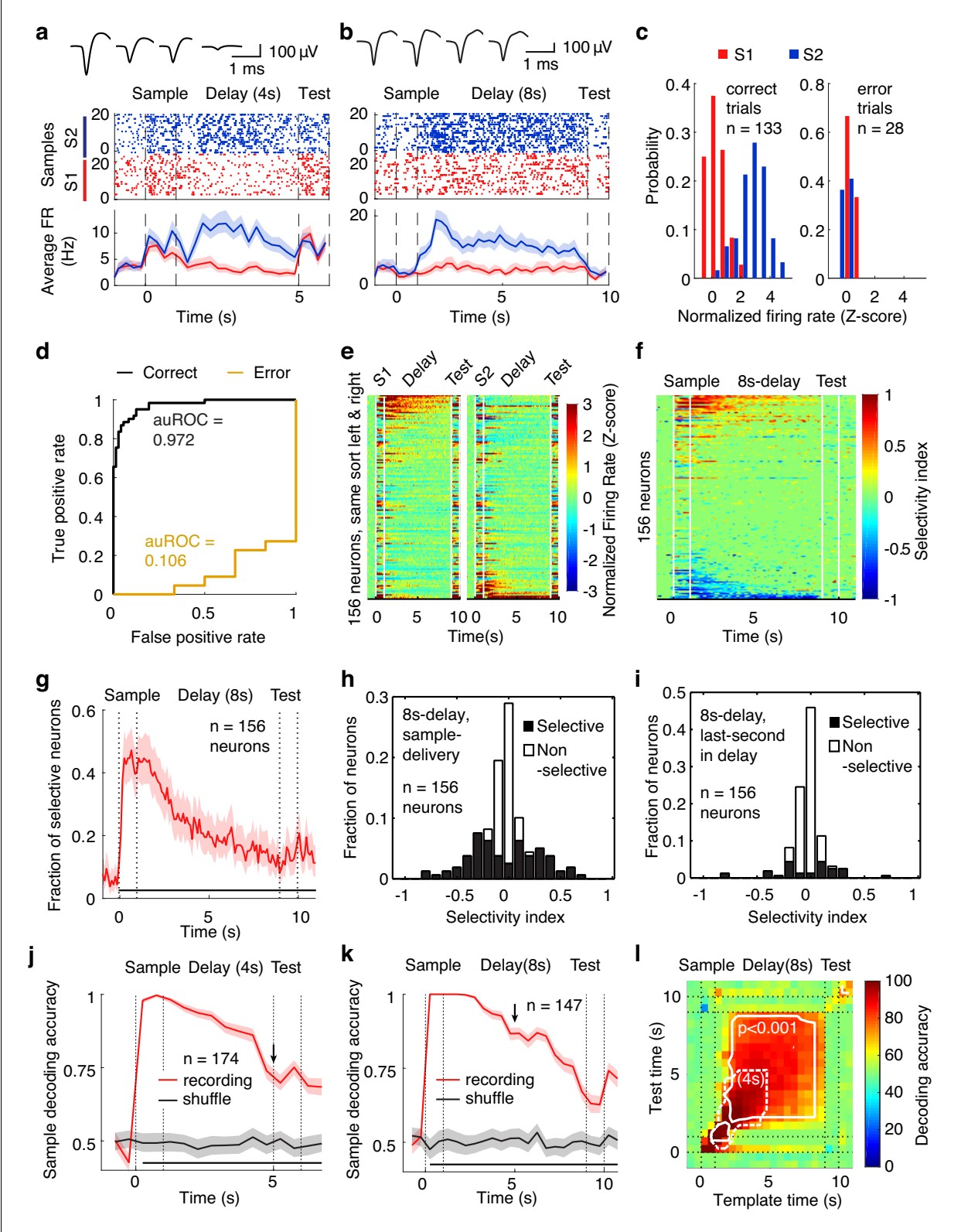

**Figure 5.** Neural correlates of the APC activity in the DNMS task. (a, b) Spike raster (top) and peristimulus time histogram (PSTH, bottom) of two example neurons recorded during the DNMS task. Shadow for PSTH shows the 95% confidence intervals. Top inset: spike waveforms from tetrode recording. The data in this and the following panels contained only correct trials, unless stated otherwise. (c) Normalized delay-period firing rate of the neuron in (a) though all the correct (left) and error (right) trials. (d) Area under receiver operating characteristic (auROC) of the neuron in (a) in correct

*Figure 5 continued on next page*

*Figure 5 continued*

and error trials. (**e**) Activity of neurons in the DNMS task with odor S1 (left) and S2 (right) as sample. Color: firing rates (FR) in Z-scores. Neurons were ordered according to the $FR^{S1} - FR^{S2}$ values during the delay period (same order in both panels). The white lines indicate the onset or offset of odor delivery. (**f**) Firing rate selectivity, defined as the firing rate following sample 1 minus that following sample 2, divided by the sum, that is $(FR^{S1}-FR^{S2})/(FR^{S1}+FR^{S2})$, for all of the recorded neurons in the DNMS task with 8 s delay duration. (**g**) Fraction of neurons with odor selectivity during the delay period, with a delay duration of 8 s. ***, p<0.001 in a chi-square test with 1 s before sample onset. Shadowing represents the 95% confidence interval of the mean from bootstrapping of 1000 repeats. (**h**) Distribution of single unit sample-odor selectivity during the sample-delivery period for the tasks of 8 s delay duration. (**i**) As in (**h**) but for the last second of the delay. (**j**) Sample decoding accuracy (resampled leave-one-trial-out cross validation accuracy) of the APC neuronal activity based on support vector machines (SVM), in the DNMS task with 4 s delay. Shadowing represents the 95% confidence interval of the mean from bootstrapping of 1000 repeats. Bottom black bar, p<0.001, two-tailed bootstrapping permutation test of 1000 repeats. (**k**) As in (**j**), for the DNMS task with 8 s delay. (**l**) Cross-temporal decoding matrix in the 8s-delay DNMS task. Dotted line, replication of the p<0.001 contour line in the 4s-delay DNMS cross-temporal decoding matrix, aligned at sample onset. See *Figure 5—figure supplements 1–4* for more details of the recording and data analysis.

DOI: https://doi.org/10.7554/eLife.43191.030

The following figure supplements are available for figure 5:

**Figure supplement 1.** Design and implementation of the op-tetrodes.

DOI: https://doi.org/10.7554/eLife.43191.031

**Figure supplement 2.** Neural correlates of the APC activity in the DNMS task.

DOI: https://doi.org/10.7554/eLife.43191.032

**Figure supplement 3.** Neural correlates of the APC activity in the DNMS task.

DOI: https://doi.org/10.7554/eLife.43191.033

**Figure supplement 4.** Selective neurons randomly distributed in different mice.

DOI: https://doi.org/10.7554/eLife.43191.034

(in 4-s delay trials) and 40% (in 8-s delay trials) of single units in the APC showed sample-odor selectivity in firing rate at the beginning of the delay period, and about 10% of single units were selective at the end of the delay period (*Figure 5g–i* and *Figure 5—figure supplement 3c–e*). Sample-selective neurons were observed in most of the mice under investigation (*Figure 5—figure supplement 4*). A population decoding analysis (bootstrap leave-one-trial-out cross-validation accuracy) based on support vector machines (SVM) also demonstrated significant single-trial decoding of sample-odor identity in APC delay activity (*Figure 5j and k*). Because the electrodes were generally positioned at different depths during the 4-s and 8-s recording sessions (estimated 200 μm apart), the difference in decoding power for the different delay durations could be due to different layers or learning experience. It is nonetheless interesting to see that the fraction of sample selective neuron in the APC was greater in the 8s-delay task than in the 4s-delay task (*Figure 5—figure supplement 3f*), both in the total fraction of selective neurons (sample-selective for more than 0.5 s) and in the subpopulation that showed longer persistent selectivity (sample-selective for more than 5 s). Furthermore, the dynamics of the decoding accuracy in the 8s-delay-tasks was not a passive continuation of the decoding accuracy in the 4 s task, as the decoding accuracy at 5 s since sample onset was much higher in the 8 s task (~85%, *Figure 5k*) than in the 4 s task (~70%, *Figure 5j*), consistent with the active recruitment of neurons for the maintenance of information in accordance with task requirements.

To further assess whether APC activity could stably maintain WM during the delay period, we performed a population cross-temporal discrimination (CTD) analysis (*Meyers et al., 2008*; *Stokes et al., 2013*). The essence of this analysis is to use the activity at a given time to decode the maintained information (the sample odor) at other time points systematically (*Figure 5l*, see Materials and methods). A higher decoding power in the off-diagonal space of a CTD plot suggests more stable information maintenance (*Meyers et al., 2008*; *Stokes et al., 2013*). In the CTD analysis of APC neuronal activity, significant decoding was observed in the off-diagonal space in tasks with both 4-s and 8 s delay periods, suggesting stable maintenance of WM information by APC neuronal activity throughout the delay period.

## Neural correlates of APC activity in the multi-sample DPA task

It has been shown that the number of stimuli used for training can bias the strategies of the animals (*Slotnick, 2001*), and this limits the generality of the results when using just two odorants as the

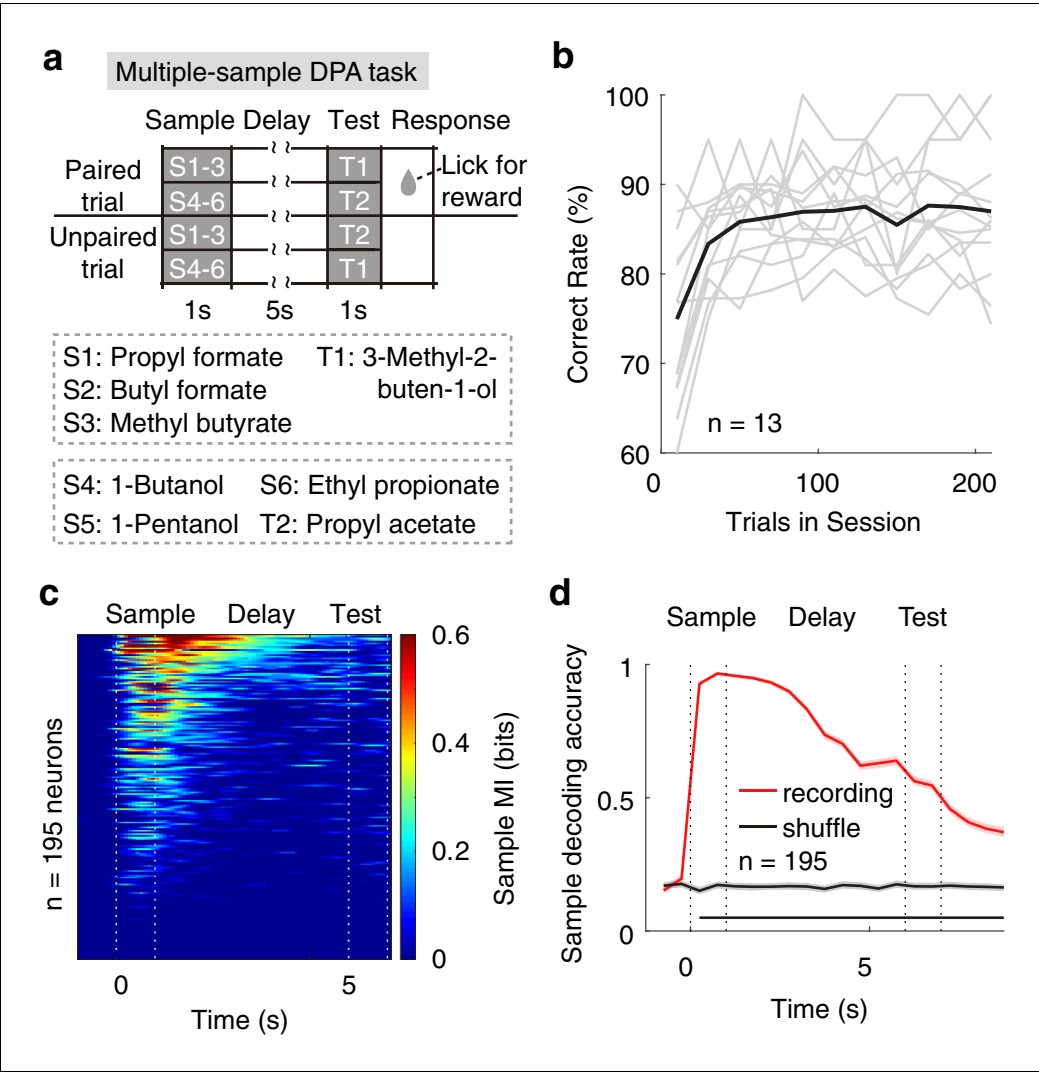

**Figure 6.** Neural correlates of APC activity in the multiple-sample DPA task. (**a**) Design of the multiple-sample DPA (MS-DPA) task. (**b**) Averaged daily performance of well-trained mice in the MS-DPA task. (**c**) Mutual information of the APC neurons for the sample odors in the MS-DPA task. Pixels were masked with p<0.001 (two-tailed permutation test of 1000 repeats against shuffled control). (**d**) Sample decoding accuracy (resampled cross validation accuracy) of the APC neurons in the MS-DPA task based on the SVM. Shadowing represent the 95% confidence interval of mean from bootstrapping of 1000 repeats. Bottom black bar, p<0.001, permutation test of 1000 repeats.

DOI: https://doi.org/10.7554/eLife.43191.035

sample odors. We therefore trained the mice with a DPA task using six odorants as samples (S1–S6) and two odorants as test (T1 and T2). In this task, the S1, S2 and S3 odors were paired with the T1 odor, whereas the S4, S5 and S6 odors were paired with the T2 odor (multiple-sample DPA, **Figure 6a**). Despite an initial relearning each day, mice performed this task well (**Figure 6b**). As for the DNMS task with two sample odors, we observed many neurons with sample-odor information during the delay period, as revealed by mutual information (MI, **Figure 6c**, see Materials and methods), which measures the degree to which responses are informative about the identity of the stimulus. A similar result was also obtained with SVM decoding analysis (**Figure 6d**).

## Neural correlates of the APC activity in the dual-task design

We then examined the APC neuronal activity in the dual task. Even after the distracting task during the delay period, APC population activity still coded the DPA samples and GNG distractors, as

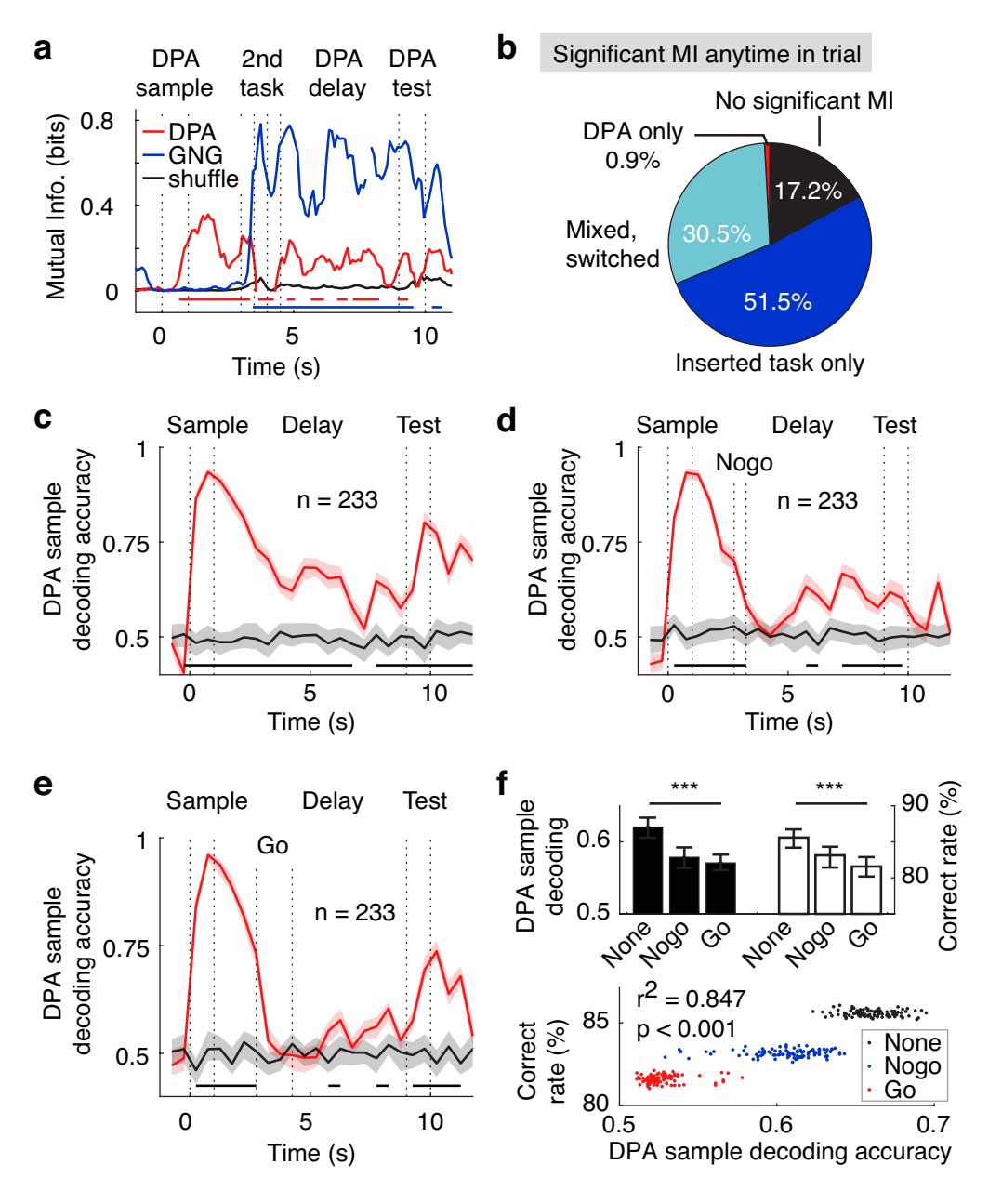

**Figure 7.** Neural correlates of the APC activity in dual-task experiments. (**a**) Example neuron with mixed DPA-sample and GNG-task odor-cue information in the dual-task. Bottom bars, p<0.001 for DPA sample (red) and GNG cue (blue), two-tailed permutation test of 1000 repeats, comparing to shuffled control. (**b**) Fraction of neurons with exclusive or mixed/switched odor-cue information in the dual-task. (**c**) DPA sample decoding accuracy (resampled cross validation accuracy) of the APC neurons in the dual-task trials without a distractor. Bottom black bar, p<0.001, permutation test of 1000 repeats, unless stated otherwise. Shadowing represents the 95% confidence interval of mean from bootstrapping of 1000 repeats. (**d**) DPA sample decoding accuracy (resampled cross validation accuracy) of the APC neurons in the dual-task trials with the No-go distractor. (**e**) Same as (**d**), but in trials with the Go distractor. (**f**) Correlations between sample decoding of APC activity and dual-task performance. Top: averaged DPA sample decoding accuracy of the late delay period (left) and correct rate in DPA performance (right) under different distractor conditions. ***, p<0.001 as determined by one-way ANOVA. Bottom: correlations of DPA sample decoding accuracy and correct rate in bootstrap resamples. r² is the Pearson correlation coefficient.

DOI: https://doi.org/10.7554/eLife.43191.036

shown in the MI analysis of one example neuron in *Figure 7a*. In the APC neuronal population, approximately 30% of neurons can code for both the DPA samples and the GNG odors (*Figure 7b*). In the SVM decoding analysis based on APC neuronal activity during the later delay period after the distracting task, one can decode the DPA sample odors in the dual-task design in the trials without distractor, with the No-go distractor, or with the Go distractor (*Figure 7c–e*). As mentioned in the previous optogenetic results, the DPA performance was related to the distractor type used in the recording sessions (*Figures 4e* and *7f*). Interestingly, we observed that the SVM decoding accuracy during the delay period after the distractor task was also related to the distractor type and to performance in the distractor task (*Figure 7f*), suggesting that APC selectivity is important for performance in the DPA task in the dual-task design.

## Discussion

Our work demonstrates that the APC is important for active maintenance in olfactory working memory. Using optogenetic manipulation in a series of behavioral tasks that temporally isolated the retention of sensory information during the delay period from decision making, we demonstrated that WM performance can be impaired in WM tasks with or without distraction. The control experiments suggested that the behavioral defects of optogenetic perturbation during the delay period may be due to impaired information maintenance. The APC population activity exhibited statistically significant mutual information and decoding power for the odor samples during the odor-delivery and delay periods. We observed that only about 10% of APC neurons carried WM information during the late delay period (*Figure 5g and i*). The small percentage of APC neurons that are involved in coding the maintained information is consistent with the sparse coding previously observed in piriform cortex (*Poo and Isaacson, 2009*; *Stettler and Axel, 2009*; *Miura et al., 2012*) (but see *Bolding and Franks, 2017*). Therefore, although the APC activity is constantly updated by ongoing sensory inputs, this sensory cortex could resist overwriting and could maintain information through population activity in the WM tasks, even when a distracting task is performed during the delay period.

### Debate about the necessity of delay-period activity in sensory cortices for WM

There is an ongoing debate concerning the necessity of delay-period activity of sensory cortices in WM. The 'essential' (*Pasternak and Greenlee, 2005*; *Scimeca et al., 2018*; *Gayet et al., 2018*) and 'unessential" (*Mendoza-Halliday et al., 2014*; *van Kerkoerle et al., 2017*; *Leavitt et al., 2017*; *Bettencourt and Xu, 2016*; *Xu, 2018*) theories argue for and against the importance of sensory cortices, respectively. Perturbation during the delay period specifically is a key experiment that can disentangle the debate. For example, in the thought experiment proposed by *Scimeca et al. (2018)*, temporal suppression of the delay-period activity of an early sensory region should produce behavioral defects if sensory regions are essential for WM maintenance. Our experimental data are consistent with this latter prediction, suggesting that a sensory region can be important for olfactory WM.

### Effect size of optogenetic suppression

We observed a small effect of optogenetic suppression on behavioral performance with shorter duration of optogenetic suppression. For example, the effect size was −0.8 and −0.68 for 5 s and 8 s delay duration, respectively, which were statistically smaller than the effect size of −1.62 observed for the 12 s delay (*Figure 1h–j*). Also, there was no significant drop in performance when the APC was suppressed in early delay period (*Figure 2b and d*). Furthermore, in the fixed optogenetic suppression varied delays task, the effects of the optogenetic suppression of APC were also quite variable (*Figure 2g*). One possibility may be that our optogenetic suppression was quite focal and spared large areas of APC, as shown by the reduction in *c-Fos* stained active neurons (*Figure 1—figure supplement 2c*). Therefore the simpler tasks with shorter-delay duration imposed less demand on APC delay-period activity, and the remaining unaffected areas within APC might be sufficient to maintain the information. Another possibility is that there could be a loop of activity involving APC and other regions (see below for more discussion). Thus, suppression of activity in other parts of the brain might contribute to our observations. Moreover, the partial optogenetic

suppression could act like a damper on recurrent activity, with a slow decay rate. This would produce the observed larger effect of the longer illumination (*Figures 1h–j* and *2b*). The observation of a smaller optogenetic effect for a short delay duration is consistent with other studies. For example, *Bolkan et al. (2017)* showed that optogenetic silencing of the medial prefrontal cortex (mPFC) impaired spatial WM performance in a task with 60 s but not 10 s delay duration.

The optogenetic silencing impairment in DPA performance is smaller than that in the DNMS task (comparing *Figure 4* with *Figure 1*). The designs of the two tasks are quite different, for example, the DPA task requires the association between sample and test odors, which is not required for the DNMS task. Therefore, the neural circuit for the DPA task might be different from that for the DNMS task, which remains to be determined in future studies.

### Distributed network interaction

The olfactory pathway downstream from the olfactory bulb is organized in a highly parallel manner, as mitral/tufted neurons in the olfactory bulb project to multiple brain regions, including the APC, the anterior olfactory nucleus, the cortical amygdala, the olfactory tubercle, and the lateral entorhinal cortex (*Ghosh et al., 2011*; *Miyamichi et al., 2011*; *Sosulski et al., 2011*; *Price and Powell, 1970*; *Davison and Ehlers, 2011*). Therefore, it was surprising that optogenetic perturbation of just one out of these five brain regions can impair olfactory WM performance, suggesting the importance for WM of the APC among the parallel olfactory pathways.

Other brain regions may contribute to the maintenance of sensory information in a WM task (as discussed in the previous section). The hypothetical existence of these regions does not conflict with the current results showing that a sensory region plays important roles in WM maintenance. The exact mechanism underlying the distributed neural circuit that underlies WM remains to be determined in the future. Furthermore, the functional role of other sensory regions (*Ghosh et al., 2011*; *Miyamichi et al., 2011*; *Sosulski et al., 2011*; *Price and Powell, 1970*; *Davison and Ehlers, 2011*) and the potential interaction between the APC and the mPFC (*Liu et al., 2014*) in the learning and well-trained phases of WM tasks remain to be determined.

### Dynamic transfer of the functional role from mPFC to APC through learning

Previously we showed that the delay-period activity of the mPFC was important during the learning but not during the well-trained phase in the olfactory WM task (*Liu et al., 2014*). Therefore, it is pertinent to examine the roles of different brain regions in learning and well-trained mice. The results of our optogenetic and recording experiments demonstrated that the APC delay activity was important in the well-trained phase. Thus, the functional role of memory maintenance may be partially transferred from the mPFC to the APC through learning. The underlying mechanisms of the transfer and the potential involvement of other brain regions remain to be determined.

### Involvement of APC in active maintenance in the dual-task WM design

The dual-task design (*Figure 4c*) explicitly challenges the central executive control ability of a subject (reviewed by *Baddeley, 2012*; *Watanabe and Funahashi, 2018*). It is an important tool to study the distributed nature of WM maintenance, which could depend on task complexity (*Christophel et al., 2017*). A simpler sensory memory task could be more dependent on a sensory cortex than a more complex task. Thus, if suppressing the delay activity of a sensory cortex in the dual task could impair WM performance, it will certainly implicate this sensory region in active information maintenance in WM. Most of the previous studies have focused on prefrontal cortex (*Watanabe and Funahashi, 2014*; *D'Esposito et al., 1995*; *Baddeley et al., 1997*). Here, we show that the delay-period activity of a sensory cortex can maintain sample information after the distracting task, and that suppressing this activity resulted in performance impairments (e.g. significantly increased false-alarm rate). Therefore the delay-period activity of this sensory cortex is involved in dual-task performance.

Neuronal activity in the APC is associated with olfactory perception, but it also varies depending on brain states, task design, or learning experience (*Courtiol and Wilson, 2017*). For this reason, the APC has long been suggested to be an associative sensory region (*Bekkers and Suzuki, 2013*; *Courtiol and Wilson, 2017*). Consistent with this notion, our results demonstrated the importance

of the APC activity beyond sensory perception. In summary, our results underscore the importance of the olfactory sensory cortex in memory maintenance beyond immediate sensory processing.

## Materials and methods

**Key resources table**

| Reagent type (species) or resource | Designation | Source or reference | Identifiers | Additional information |
|---|---|---|---|---|
| Strain, strain background (*Mus musculus*) | VGAT-CHR2:B6.Cg-Tg(*Slc32a1*-COP4*H134R/EYFP)8Gfng/J | Dr. Guoping Feng (*Zhao et al., 2011*) | VGAT-ChR2-EYFP line 8 | |
| Antibody | goat polyclonal anti-GFP (FITC) | Abcam | RRID: AB_305635 | (1:200) |
| Antibody | rabbit polyclonal anti-*c-Fos* | Synaptic Systems | RRID: AB_2231974 | (1:1000) |
| Antibody | Cy5 goat polyclonal anti-rabbit | Jackson | RRID: AB_2338013 | (1:2000) |
| Chemical compound, drug | DAPI | Beyotime | catalog number: C1002 | |
| Chemical compound, drug | propyl formate | Sigma-Aldrich | catalog number: 245852 | |
| Chemical compound, drug | butyl formate | Sigma-Aldrich | catalog number: 261521 | |
| Chemical compound, drug | methyl butyrate | Sigma-Aldrich | catalog number: 246093 | |
| Chemical compound, drug | 3-methyl-2-buten-1-ol | Sigma-Aldrich | catalog number: W364703 | |
| Chemical compound, drug | 1-butanol | Sigma-Aldrich | catalog number: B7906 | |
| Chemical compound, drug | 1-pentanol | Sigma-Aldrich | catalog number: 398268 | |
| Chemical compound, drug | ethyl propionate | Sigma-Aldrich | catalog number: 112305 | |
| Chemical compound, drug | propyl acetate | Sigma-Aldrich | catalog number: 133108 | |
| Software, algorithm | MATLAB | MathWorks | RRID: SCR_001622 | |
| Software, algorithm | LIBSVM | https://www.csie.ntu.edu.tw/~cjlin/libsvm/ | RRID: SCR_010243 | |

### Animals

All experiments were performed in compliance with the animal care standards set by the U.S. National Institutes of Health and have been approved by the Institutional Animal Care and Use Committee of the Institute of Neuroscience, Chinese Academy of Sciences (Shanghai, China). B6.Cg-Tg (*Slc32a1*-COP4*H134R/EYFP)8Gfng/J mice, commonly referred to as VGAT-ChR2 mice, were used in optogenetic experiments, and littermates of the same sex were used as controls. All mice were healthy male, group housed, of age 8–12 weeks and 20–30 g in weight at the start of training. For all experiments, individual animals are presented as individual data points, otherwise the sample sizes (n) are shown in the corresponding figures. In each experiment condition, individual data points represent individual animals (biological replicates). Data from the same animal across multiple conditions are indicated by color or joint line segments and treated accordingly (e.g., statistical tests with repeated measures). When bootstrap resampling methods were used (technical replicates), this is clearly stated along with the number of resampling repeats. We ensured that each group in the behavior studies included at least 10 mice, which had been shown to be sufficient to detect the effects of optogenetic manipulations in comparable working memory tasks (*Liu et al., 2014*).

## Behavioral setups

Some of the following methods are similar to those previously published (*Liu et al., 2014*; *Han et al., 2018*). We utilized the DNMS task, DPA task and dual-task as described previously (*Liu et al., 2014*; *Han et al., 2018*) or as described below. In brief, the olfactometry apparatus was enclosed in sound-proof training boxes. An embedded system, custom built around a PIC Digital Signal Controller (dsPIC30F6010A, Microchip, Chandler, AZ), was used to control the olfactory cue and water delivery by switching solenoid valves and to detect lick responses with an infra-red beam break detector or a capacitance sensor. The air-and-odorant-mixture nozzle/lick-port/beam-breaker assembly were 3D-printed and placed in front of the mouse. The air flow rate was controlled at 1.35 L/min during and between odor deliveries. Propyl formate (245852, Sigma-Aldrich, St. Louis, MO), butyl formate (261521, Sigma-Aldrich), methyl butyrate (246093, Sigma-Aldrich), 3-methyl-2-buten-1-ol (W364703, Sigma-Aldrich), 1-butanol (B7906, Sigma-Aldrich), 1-pentanol (398268, Sigma-Aldrich), ethyl propionate (112305, Sigma-Aldrich) and propyl acetate (133108, Sigma-Aldrich) were used at 1:500 concentration in mineral oil (v/v; O1224, Fisher Scientific, Pittsburgh, PA), or stock odorants are kept and evaporated in an air-tight bottle. During odor delivery, the odorant vapor was exposed to individually controlled air flow, then mixed with air at 1:10 concentration (v/v). The concentration of odors was measured using a photoionization detector (200B miniPID, Aurora Scientific Inc, St., Aurora, Canada), and the concentration during the delay fell to the baseline level within 1 s after valve shut-off. Behavior events and timings were simultaneously sent to and recorded by a computer using customized software written in Java (Oracle, Redwood Shores, CA).

## Behavior training

The DNMS task was carried out as described previously (*Liu et al., 2014*). Briefly, a sample olfactory stimulus was presented for 1 s at the start of a trial, followed by a delay duration of 4–40 s (mice need to retain the information of the first stimulus (sample) during the delay duration), then a test olfactory stimulus for 1 s, identical to (in match trials) or different from (in non-matched trials) the sample. After a 1-s pre-response-delay, mice were trained to lick in the 0.5 s response window only in non-matched trials. *Hit* or *False alarm* was defined as detection of lick events in the response window in a non-match or match trial, respectively. Similarly, *Miss* or *Correct rejection* were defined as the absence of a lick event in the response window in a match or non-match trial, respectively. A reward of 5 µl water was triggered immediately only after *Hit*; mice were neither rewarded nor punished following other responses. Mice were allowed to perform up to 300 trials each day, a consecutive combination of 10 miss and correct rejection trials also triggered the end of the session. Only the trials within well-trained performance windows (no less than 80% correct rate within consecutive 40 unperturbed trials) were included in the data analysis, unless stated otherwise. In the fixed delay, varied optogenetic stimulation tasks, the well-trained window was reduced to 20 consecutive unperturbed trials because of the reduced number of unperturbed trials in each block. In the fixed optogenetic stimulation, varied delay task, the well-trained window is defined as no fewer than six correct trials in eight unperturbed trials in a 32-trial block. In the increasing delay duration experiment, the mice were trained to perform the DNMS task with 5 s delay to the well-trained criterion, then the delay duration was increased every day; the first 100 trials were included in the the analysis to accomplish direct parallel comparison. An inter-trial interval twice as long as the delay duration separated consecutive trials.

In the DPA task, one of two sample odors was presented for 1 s at the start of a trial, followed by a delay duration of 13 s, then one of two different test odors was presented for 1 s. One of the sample odors and one of the test odors formed a rewarded pair, while the other two odors formed another rewarded pair. After a 1-s pre-response-delay, mice were trained to lick in the 0.5-s response window only in paired trials. *Hit* or *False alarm* was defined as the detection of lick events in the response window in a paired or non-paired trial, respectively. Similarly, *Miss* or *Correct rejection* were defined as absence of a lick event in the response window in a paired or non-paired trial, respectively. A reward of 5 µl water was triggered immediately only after *Hit* responses; mice were neither rewarded nor punished following other responses. Mice were allowed to perform up to 300 trials each day, a consecutive combination of 10 miss and correct rejection trials also triggered the end of the session. Only the trials within well-trained performance windows (no fewer than 80%

correct rate within consecutive 40 unperturbed trials) were included in the data analysis, unless stated otherwise. An inter-trial interval as long as the delay durations separated consecutive trials.

The multiple sample DPA (MS-DPA) task is like the DPA task described previously, except that the number of candidate sample odors was increased to six; three of the samples and one test formed rewarded pairs (e.g., S1, S2, S3 and T1), and the other three samples and the other test formed the remaining rewarded pairs (e.g., S4, S5, S6 and T2). The delay duration in the MS-DPA task is 5 s.

In the dual-task, a secondary Go/No-go task was inserted into the delay duration of the DPA task. Delivery of the olfactory cue for the Go/No-go task started 3 s into the DPA task delay period and continued for 0.5 s. After a 0.5 s pre-response-delay, mice were trained to lick in the 0.5-s response window after the Go stimulus. *Hit* or *False alarm* of the Go/No-go task was defined as the detection of lick events in the response window in a Go or No-go trial, respectively. Similarly, *Miss* or *Correct rejection* were defined as absence of a lick event in the response window in a Go or No-go trial. A reward of 5 µl water was triggered immediately only after *Hit* responses; mice were neither rewarded nor punished following other responses. The sample and test stimuli in the DPA task and the stimulus in the Go/No-go task were arranged independently in a pseudo-random fashion. Mice were allowed to perform up to 288 dual-task trials each day, a consecutive combination of 10 miss and correct rejection trials also triggered the end of the session. For the well-trained phase studies, only the trials within well-trained performance windows (no fewer than 80% DPA correct rate within consecutive 40 unperturbed trials) were used for data analysis; for the learning phase, all trials were included in the analysis, unless stated otherwise. An inter-trial interval as long as the delay duration of the DPA task separated consecutive trials. The Go/No-go task is omitted in one third of all trials. The performance in the 4th day of the dual-task learning stage was used to estimate dual-task interference (*Figure 4f*).

Mice were water restricted for 48 hr before the start of training, followed by habituation, shaping and learning phases, before well-trained optogenetic sessions. In the habituation phase, mice were head-fixed in the olfactometry apparatus for 2 hr and allowed to lick water from the water port, encouraged by program-controlled water delivery. Typically in 1 to 2 days, mice could learn to trigger more than 100 water rewards spontaneously. In the shaping phase, only non-match or paired trials were presented, and water was pseudo-randomly and programmatically delivered after 1/3 of the *Miss* responses to encourage task engagement. The shaping phase ended once the mice could perform more than 32 *Hit* responses in 40 consecutive trials (80% performance). In the learning phase, both match and non-match, paired and non-paired trials were presented, and each four consecutive trials that consisted of four balanced sample-test-reward combinations was shuffled. The learning phase ended when the mice were able to perform more than 32 *Hit/Correct Reject* responses in 40 consecutive trials (80% performance). The performance (referred to as the 'correct rate' in the labels of figures) was defined as the total fraction of hit and correct rejection responses. The sensitivity index (*d'*) was defined as *d'=norminv (Hit rate) – norminv (False choice rate)*. The lick efficiency was defined as the number of rewarded licks divided by the sum of rewarded and unrewarded licks in the response window. (*Liu et al., 2014*).

For all experiments with blind design, RQ Hou, HM Fan and ZQ Chen labeled the mice with unique numbers without revealing the genotype; they did not participate in the behavior or optogenetic experiments. The genotype of mice would only be revealed after the experiments and statistical analysis of individual mice had been finished. Furthermore, all of the mice participated in the same behavior studies in identical sequences in the task design, so there is no need for further randomization of samples in this study.

## Stereotaxic implantation of optical fiber

Implantation of optical fiber was carried out in a manner similar to that described previously (*Liu et al., 2014*). Briefly, mice were anaesthetized and placed in a stereotaxic instrument (Stoelting Co. Wood Dale, IL). After removal of the scalp, periosteum and other associated soft tissues, a custom designed steel plate was fixed onto the skull using tissue adhesive (1469 SB, 3M, Maplewood, MN) and dental cement. Craniotomies of roughly 1 mm in diameter were made bilaterally above the anterior piriform cortex (APC). For optical fiber implantation, two optic fibers (200 µm in diameter, Leizhao Biotech., Shanghai, China) with ceramic ferrule were implanted at A.P. 1.78 mm, M.L. 2.60 mm and D.V. 3.40 mm. A thin layer of silicone elastomer (Kwik-Sil, WPI, Sarasota, FL) was applied to

protect the brain tissue, then dental cement was applied to connect the skull, plate and optical fibers for structural support. Antibiotic drug (ampicillin sodium) was injected i.p. for three consecutive days after surgery.

## Optogenetic experiments

Optogenetic experiments were performed as described previously (*Liu et al., 2014*). Briefly, an optical patch cable (200 μm in diameter, N.A. 0.37) was used to connect the implanted optic fiber (through a ceramic sleeve) to a laser source (BL473T3-50FC, SLOC, Shanghai, China). Laser power was programmatically controlled by a PIC microcontroller through analog voltage input. Output laser power was measured with a laser power meter (LP1, Sanwa Electric Instrument Co., Tokyo, Japan) and compensated for optical power loss in fiber implant and coupling attenuation (determined before surgery). In optogenetic experiments with laser power ramp, the dynamics were further calibrated by an oscilloscope.

In the DNMS task optogenetic sessions (*Figures 1h–1n* and *2b–2d*, and *Figure 1—figure supplement 5*), the optogenetic suppression was arranged in an interleaved one-trial-on, one-trial-off fashion; each session started with a laser-off trial. In the fixed delay, varied optogenetic stimulation tasks, various optogenetic stimulation designs, including laser-off trials, were pseudo-randomly carried out on a trial-by-trial basis. In the DPA task and dual-task optogenetic sessions (*Figure 4b,g,h* and *Figure 4—figure supplement 1*), the optogenetic suppression was arranged in an interleaved one-block-on, one-block-off fashion, and each block consisted of 24 trials; each session started with a laser-off block.

For the *c-Fos* activity labeling, three mice of the same VGAT-ChR2 genotype were optogenetically stimulated in 3-sec-on, 17-sec-off cycles for 50 trials, mimicking the DNMS task optogenetic stimulation, while no olfactory cues were presented to reduce background noise.

## Immunostaining and imaging

After the conclusion of experiments, mice were deeply anesthetized with sodium pentobarbital (120 mg/kg) and then perfused transcardially with 20 mL 0.9% NaCl solution followed by 20 mL paraformaldehyde PBS solution (PFA, 4%, w/v). The brains were fixed in PFA solution overnight and sectioned. Slices were washed with PBS, fluorescence-enhanced with the anti-GFP antibody (FITC, ab6662, Abcam, Cambridge, MA) or *c-Fos* antibody (226003, Synaptic Systems, Germany) then Cy5 goat-anti-rabbit IgG (111-175-144, Jackson), incubated with DAPI (C1002, 1:1000, Beyotime, Shanghai, China), and later mounted with DAPI. Fluorescence images were then obtained with an epifluorescence microscope (Eclipse 80i, Nikon, Tokyo, Japan) with a 10X (CFI Plan Apo Lambda, 0.45 N.A.) or a 20X (CFI Plan Apo Lambda, 0.75 N.A.) objective lens, and were analyzed with ImageJ software (NIH, U.S.).

## Surgical implantation of the (op-)tetrode microdrive

The implantation procedure was similar to that reported previously (*Liu et al., 2014*). Briefly, all surgical tools and the microdrive were sterilized by ultraviolet radiation for more than 20 min before implantation. One cranial window of 0.5 by 1 mm was made in either hemisphere centered at A. P. +1.42 mm and M.L. 2.5 mm, and electrodes were lowered at D.V. 3 mm to target the APC. Tissue gel (1469 SB, 3M) and dental cement were carefully applied to cover the exposed brain tissue and to fix the microdrive. Antibiotics (ampicillin sodium, 20 mg/mL, 160 mg/kg b.w.) was injected for three consecutive days after surgery. Behavior training started 7 to 14 days after that. Recordings of behaving mice were made with electrodes lowered for approximately 50 μm each day after the last day of shaping.

## Electrophysiology recording

All recorded neurons were single units. Wide band signals (0.5–8000 Hz) from all tetrodes were amplified (×20,000) and digitized at 40 kHz with the Multichannel Acquisition Processor (Plexon Inc, Dallas, Texas) and all data were saved to hard-disks. Detection and sorting of spike events were performed offline as described in a previous paper (*Liu et al., 2014*). Briefly, offline spike detection was performed with Offline Sorter (Plexon Inc). Raw signals were filtered in 250~8000 Hz to remove field potentials. Signals larger than five times the standard deviation recorded on any

recording site of the tetrode were considered to be spike events. Principle component analysis (PCA) was performed for tetrode-waveforms to extract the first three principle components (PCs) explaining the largest variance. Then, the contour or valley-seek clustering techniques provided by Offline Sorter were performed in 2D or 3D feature space (including principle components) of wave-forms. Single units were included only if no more than 0.15% of the spikes occurred within a 2 ms refractory period (false-alarm rate) and the averaged firing rate was higher than 2 Hz. All of the neurons that we sorted exhibited large signal-noise ratios and small false-alarm rates, demonstrating the high quality of single-unit sorting (*Figure 1—figure supplement 3*). All recording sites were further confirmed by passing current (50μA, 100 ms, 1 Hz x five pulses) through the electrodes, and verified with DAPI staining and immunochemistry 1 day after the lesion (*Figure 5—figure supplement 1c*).

## Quantification and statistical analysis

All further analyses were performed by custom-written codes in MATLAB (Mathworks, Natick, MA) and Java (Oracle). Statistical significance was defined as $p < 0.05$ unless stated otherwise.

## Optogenetic suppression

Each point in the figures represents the averaged result from one mouse. Therefore the data from one mouse were never tested multiple times in one experiment. The genotype of each mouse would only be revealed after the experiments and statistical analysis of individual mice had been finished. Then the behavior performance in trials without and with optogenetic suppression were calculated separately for the experiment group and the control group. For the optogenetic tasks, mixed-between-within-ANOVA were performed, in which 'between-group' factors were defined by genotype and 'within-group' factors were defined as laser-on and -off trials (*Liu et al., 2014*) (*ranova* function in MATLAB). Statistical significance was defined by the genotype and laser on-off interaction p-value. One-way ANOVA (MATLAB function *anovan*) of correct rate change (Δ correct rate) was performed for ChR2 mice across all delay durations.

For the permutation test, the absolute difference between the means of two groups of samples was calculated, then the two groups were pooled together. For a number (usually 1000) of repeated permutations, the pooled data were regrouped into two subsets that matched the original two groups in size, then the absolute differences between the means of the subsets were calculated and stored in a vector. The p value is the probability that the permuted difference is equal to or larger than the test group difference. The paired permutation test was similar to the permutation test, but the test data were mean differences between data pairs, and the sign (positive or negative) was randomly flipped in each repeat of the permutation.

For the detection of outliers, the MATLAB function *isoutlier* was used with two built-in methods, 'quantile', and 'grubbs'.

## General linear model

The WM task trials can be defined by a combination of task parameter values, including sample odor, test odor, match or non-match relationship, genotype of the mice, laser-on or laser-off per trial, delay duration (transformed to an exponential time-decay value based on *Figure 1d*), and the perturbation design (no perturbation, perturb baseline or perturb delay). A subset of the aforementioned task parameters, expressed as vector $X$, and a coefficient vector $b$, defined a linear combination $X^T b$. The trials corresponding to the combination of X were grouped together, and the averaged correct rate of the trials was determined as response value $\mu$. The model is $\mu = X^T b$. A series of seven models were systematically generated as described in *Figure 3a*, and fitted with MATLAB function *fitglm()*. Through bootstrapping of 500 repeats (resampling the corresponding trials with replacement), the r-squared and Akaike information criterion (AIC) values for each model were determined, and the model (#6) that had the lowest AIC value was chosen for further analysis. The combination of each $X$ and $b$ in the model provided the numerical relationship between a task parameter and the averaged change in correct rate in *Figure 3d*. For *Figure 3c* and *Figure 3—source data 3,4*, the coefficients in the model were calculated for all corresponding trials, without resampling.

## Spike count, auROC and activity heat map

Averaged firing rates of individual neurons in well-trained correct trials (between 33 and 160 trials, depending on the specific session) for different olfactory sample were binned into 200 ms periods (*Figure 5e* and *Figure 5—figure supplement 3a*). The baseline period was defined as the 1 s before the onset of the sample odor delivery. Firing rates from baseline of each trial were averaged to form the baseline activity vectors for each neuron. The mean and standard deviation of this baseline activity vector were used to convert averaged firing rates into Z-scores. auROC of all the neurons were calculated with the normalized firing rates using the *perfcurve* function in MATLAB. The activities of all neurons following different sample odors were sorted by the differences between sample odors during the delay period, and plotted as a heat map using the 'jet' color-map defined in MATLAB.

Firing rate selectivity (*Figure 5d* and *Figure 5—figure supplement 3b*) was defined as the firing rate following sample 1 minus that following sample 2, divided by the sum of these firing rates, for example $(FR^{S1}-FR^{S2})/(FR^{S1}+FR^{S2})$, which resulted in a $[-1,1]$ range. The results were plotted as a heat map using the 'jet' color-map defined in MATLAB.

## Mutual information

Briefly, the spike counts for each neuron during well-trained correct-task trials were binned into 500 ms windows moving at 100 ms resolution, and transformed into firing rates, then separated according to the sample odor. The distribution of firing rates within a time window among all corresponding trials for the same sample odor were fitted to a Gaussian distribution with the MATLAB *fitdist()* function. The probability of a response rate r, *P[r]*, was defined by the probability density function of the fitted Gaussian distribution. The probability of a stimulus *P[s]* was defined as the fraction of trials with sample s in all the trials. The conditional probability *P[r|s]* represented the probability of a response rate r given that sample was s. The mutual information for the sample of a neuron in a task was thus calculated as:

$$\mathrm{MI} = \sum_{s} P[s] \int dr P[r|s] log_2 \left( \frac{P[r|s]}{P[r]} \right)$$

## SVM decoding

The population decoding analysis was performed with the LIBSVM library (https://www.csie.ntu.edu.tw/~cjlin/libsvm) according to the library's documented instructions. Briefly, the spike counts for each neuron during well-trained correct task trials were binned into 500 ms windows and grouped by sample odors. Only neurons with more than 30 well-trained trials for each sample odor were included to minimize overfitting. The firing rates for all neurons were normalized to the [0, 1] range. For each repeat, the trials were first randomly partitioned to a training set (no fewer than 30 trials) and a testing set (one trial for each sample odor, unless stated otherwise), with no overlap between the training and testing sets. From the training set, the firing rates of all neurons in 30 bootstrap trials for each sample odor were selected to train the support vector machine (SVM) with the *radial basis function* (RBF) kernel. From the testing set, one trial for each sample odor was used to test the classification accuracy. All trials were then randomly re-partitioned and the training and testing repeated. Averaged decoding accuracy was obtained with 500 repeats. Because the training and testing sets never overlapped and all trials were likely to be used both in training set and testing set due to the large repetition number, the average decoding accuracy equals the cross validation accuracy in a resampled leave-one-trial-out form. A good combination of the two parameters for the RBF kernel, c and $\gamma$, were grid-searched using exponentially growing sequences in the ranges $2\hat{}[-5, 5]$ and $2\hat{}[-10, 0]$, respectively. For the shuffled control, the procedure was repeated 1000 times with the nominal sample odor for each trial randomly redistributed. For the cross temporal decoding analysis, the c and $\gamma$ values were kept at the previous obtained value, and the decoding accuracy was calculated using template and test activity vectors from different time windows across the entire length of a trial. For the study of correlation of behavioral performance and population neuronal decoding (*Figure 7f*), the decoding accuracy was obtained using averaged firing rates from 1 s before the test onset to the test offset, to reflect the neuronal activity that is important to decision making in the DPA task.

## Data and software availability

### Data availability

All data generated or analyzed during this study can be found at https://dx.doi.org/10.5061/dryad.dt5h4m1. Source data files have been provided for *Figures 1–4*.

### Code availability

The hardware design and software source code for the automatic training system (*Han et al., 2018*) and the custom-written computer codes that support the findings of this study can be obtained at Github (*Zhang, 2019*) (https://github.com/wwweagle/; copies archived at https://github.com/elifesciences-publications/BehaviorParser; https://github.com/elifesciences-publications/zmat; https://github.com/elifesciences-publications/ephysParser; https://github.com/elifesciences-publications/spk2fr; https://github.com/elifesciences-publications/PIC20Odor and https://github.com/elifesciences-publications/SerialJ).

# Acknowledgements

We thank Dr. MP Stryker, B Richmond and R Desimone for communication on head-fixed mice preparation and behavioral design; Dr. GP Feng for VGAT-ChR2 mice; Dr. G Laurent for communication on population analysis; Dr. LP Wang for help with optogenetics; Dr. LN Lin for help in tetrode drive manufacturing; Dr. Y Dan and JF Erlich for communication on spike sorting; Drs. ZL Qiu, MM Luo, HL Hu and X Yu for help on transgenic mice; Dr. AK Guo for help in PID experiments; the Imaging facility of ION and Dr. Q Hu for help with imaging experiments; Dr. YF Li for drawing the summarizing schematic; and Dr. MM Poo for critical comments on the manuscript. The work was supported by the National Science Foundation for Distinguished Young Scholars of China (31525010, to CTL), the Shanghai Municipal Science and Technology Major Project (Grant No. 2018SHZDZX05), the Strategic Priority Research Program of the Chinese Academy of Sciences (Grant No. XDB32010100), the General Program of the Chinese National Science Foundation (31471049), the Instrument Developing Project of the Chinese Academy of Sciences (Grant No. YZ201540), the Key Research Program of Frontier Sciences of the Chinese Academy Sciences (QYZDB-SSW-SMC009), the Key Project of the Shanghai Science and Technology Commission (No.15JC1400102, 16JC1400101), the China–Netherlands CAS-NWO Programme: The Future of Brain and Cognition (153D31KYSB20160106), the Spanish Ministry of Science, Innovation and Universities and the European Regional Development Fund (BFU2015-65318-R, to AC), and the CERCA Programme/Generalitat de Catalunya (to AC).

# Additional information

### Funding

| Funder | Grant reference number | Author |
| --- | --- | --- |
| National Natural Science Foundation of China | Distinguished Young Scholars of China (31525010) | Chengyu T Li |
| Chinese Academy of Agricultural Sciences | Instrument Developing Project YZ201540 | Chengyu T Li |
| Shanghai Municipal Science and Technology Commission | 15JC1400102 | Chengyu T Li |
| Ministerio de Ciencia, Innovación y Universidades | | Albert Compte |
| European Regional Development Fund | BFU2015-65318-R | Albert Compte |
| CERCA Programme/Generalitat de Catalunya | | Albert Compte |
| Shanghai Municipal Science and Technology Commission | 2018SHZDZX05 | Chengyu T Li |

| National Natural Science Foundation of China | General Program 31471049 | Chengyu T Li |
| Shanghai Municipal Science and Technology Commission | 16JC1400101 | Chengyu T Li |
| Chinese Academy of Agricultural Sciences | Key Research Program of Frontier Sciences QYZDB-SSW-SMC009 | Chengyu T Li |

The funders had no role in study design, data collection and interpretation, or the decision to submit the work for publication.

## Author contributions

Xiaoxing Zhang, Data curation, Software, Formal analysis, Validation, Visualization, Methodology, Writing—original draft, Writing—review and editing; Wenjun Yan, Conceptualization, Data curation, Software, Formal analysis, Visualization, Methodology; Wenliang Wang, Data curation, Formal analysis, Visualization; Hongmei Fan, Data curation, Project administration; Ruiqing Hou, Yulei Chen, Zhaoqin Chen, Data curation; Chaofan Ge, Data curation, Formal analysis; Shumin Duan, Conceptualization, Funding acquisition, Project administration; Albert Compte, Conceptualization, Formal analysis, Methodology, Writing—original draft, Writing—review and editing; Chengyu T Li, Conceptualization, Formal analysis, Supervision, Funding acquisition, Visualization, Methodology, Writing—original draft, Project administration, Writing—review and editing

## Author ORCIDs

Xiaoxing Zhang https://orcid.org/0000-0001-5229-6091
Chengyu T Li https://orcid.org/0000-0001-6829-0209

## Ethics

Animal experimentation: All experiments were performed in compliance with the animal care standards set by the U.S. National Institutes of Health and have been approved by the Institutional Animal Care and Use Committee of the Institute of Neuroscience, Chinese Academy of Sciences (ION-2018010).

## Decision letter and Author response

Decision letter https://doi.org/10.7554/eLife.43191.041
Author response https://doi.org/10.7554/eLife.43191.042

## Additional files

### Supplementary files

• Transparent reporting form
DOI: https://doi.org/10.7554/eLife.43191.037

### Data availability

All data generated or analyzed during this study are available on Dryad under doi:10.5061/dryad.dt5h4m1. Source data files have been provided for Figures 1-4.

The following dataset was generated:

| Author(s) | Year | Dataset title | Dataset URL | Database and Identifier |
|---|---|---|---|---|
| Zhang X, Yan W, Wang W, Fan H, Hou R, Chen Y, Chen Z, Duan S, Compte A, Li CT | 2019 | Data from: Active information maintenance in working memory by a sensory cortex | https://dx.doi.org/10.5061/dryad.dt5h4m1 | Dryad Digital Repository, 10.5061/dryad.dt5h4m1 |

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
