## [Decision Letter]

Thank you for submitting your article "Active information maintenance in working memory by a sensory cortex" for consideration by *eLife*. Your article has been reviewed by three peer reviewers, and the evaluation has been overseen by a Reviewing Editor and Catherine Dulac as the Senior Editor. The following individuals involved in review of your submission have agreed to reveal their identity: Kartik Sreenivasan (Reviewer #2).

The reviewers have discussed the reviews with one another and the Reviewing Editor has drafted this decision to help you prepare a revised submission.

Summary:

This is a thorough study in which the authors ask if sensory cortices play a key role in working memory. The authors approach the question using behavior, single-unit recordings, and optogenetic interventions. The reviewers and I agreed that the findings were potentially exciting if they could be interpreted clearly. However, there were some critical concerns that the authors must address, including some clarifying experiments:

Essential revisions:

1) The authors must show the effect size of their optogenetic manipulation, as without this one cannot evaluate the results. Reviewer 2 has a detailed account of the requirements for this analysis.

2) The authors must disentangle the effect of duration of stimulation vs. delay length. The reviewers ask that the authors carry out additional experiments in control and experimental protocols, using the DNMS task, in which they compare the effect of optogenetic stimulation of short 3s vs. long 12s duration, while keeping the delay the same.

3) The authors should address numerous points raised by the reviewers to better explain, clarify, and provide a rationale for their data. For example, they should address reviewer's concerns on whether the light is affecting other regions of the brain. This and numerous other points of clarification have been raised by the reviewers.

4) Please note that *eLife* uses Figure Supplements (supplements to specific figures) but not a separate supplemental section. When possible, please add supplemental material to the appropriate figures., as is commonly done in *eLife* papers

*Reviewer #1:*

This is a thorough study in which the authors ask if sensory cortices play a key role in working memory. The authors utilize a variety of behavioral tasks to test olfactory working memory, intervene with optogenetics in the APC, and do single-unit recordings from the APC.

The behaviour is convincing and the technical details seem to be carefully handled. For example, the authors use PIDs to show clean removal of odor between stimuli. I appreciate the care taken to try out a range of WM tasks including a difficult one where there is an intervening distractor task during the delay period of a delayed paired association. There is a clear effect of the optogenetic stimulation in each case.

I have a concern about how well the ChR2 optical stimulus might have been localized. The authors use an optical fiber so most of the light will indeed be directed just below the fiber end, into the APC. Nevertheless, there may be significant scatter. In Figure 1E and Figure 1—figure supplement 2 we see that expression of VGAT-ChR2 is rather broad, and there are several regions which might be affected. With this in mind, I wonder if the authors may wish to revisit their statement in the Discussion that only one out of 5 potential regions were perturbed. I worry that the core conclusion, that the sensory cortex is primarily involved in the working memory, may be weakened if several regions are affected by light scatter. One way to address this might be to obtain recordings from other relevant regions in the olfactory pathway, though I do understand that this is a substantial effort. Another possibility might be *c-Fos* activity labelling.

The authors also perform single unit recordings and demonstrate sustained activity through the delay period in about 10% of neurons. While this seems reasonably clear, it would be nice to have a more complete analysis here. How many do this on single-trials? Is there more we might understand from a single-trial analysis? What might we infer from a closer comparison of 4-s vs. 8-s delay recordings?

*Reviewer #2:*

This work addresses a critical limitation of so-called sensory recruitment model – namely that there is no causal evidence that sensory delay activity is involved in WM. This paper is timely, the experiments are thorough and address nearly all potential objections, and the results are clear. Overall, this is an excellent paper. However, there are a few concerns that the authors need to address that could potentially affect the interpretation of the results. Beyond that, the clarity in presentation of the results, particularly when it comes to the organization of the figures, is lacking and needs quite a bit of work.

1) The authors do not include any measures of effect size, indicate the main effects of the ANOVA, or provide sufficient information for non-significant ANOVAs. These all hinder the ability to evaluate the results. For example, the missing effect sizes don't allow for a proper comparison between the main analyses and the control analyses (e.g., Figure 1 vs. Figure 2). Additionally, it seems possible that the light on vs. off had an effect on control mice as well (albeit a smaller one). If this were true, it would require further discussion.

2) The authors need to address one possible alternate interpretation of their results. Could it be the case that APC perturbation disrupted the delay activity of downstream neurons (e.g., in PFC) that are critical for the memory? How are these results inconsistent with that view?

3) The authors' claim rests on the fact that light suppressed the activity of APC neurons in the VGAT-ChR2 mice. However, as shown in Figure 1F and Figure 1—figure supplement 4, there are a variety of responses to the light stimulus. In order to support their claims, the authors need to address this issue directly. For example, a statement clarifying that a wide majority of neurons demonstrated optogenetic suppression, or a simple analysis demonstrating that more neurons were suppressed that would be expected by chance would help advance the claim that the net effect of the manipulation was suppression.

*Reviewer #3:*

In their manuscript, Active information maintenance in working memory by a sensory cortex, Zhang and colleagues propose to test whether activity in the sensory cortex (in particular the anterior piriform cortex, APC) is required in maintaining working memory (essential theory) by engaging mice in olfactory behaviors, and further using optogenetic manipulations and the monitoring neural correlates of these behaviors at the level of APC. The authors train head-fixed mice in several olfactory tasks: a delayed non-match to sample (DNMS) task, a (multi-sample) delayed paired association (DPA) task, and a dual version of the DPA task, where a distracting go-no-go (GNG) task is added in the delay period. They further perform controls aimed at determining whether optogenetic suppression of APC activity impairs sensory perception of olfactory stimuli and screen candidate GLMs to identify a good fit for the observed behavioral performance. In view of their optogenetic manipulations and APC electrophysiological recordings, the authors conclude that sensory cortex (APC) is critical in the active maintenance of olfactory working memory.

Understanding the contribution of sensory cortex in supporting working memory is important, and the authors have performed numerous experiments and analyses designed to get at this question from multiple angles. Yet, I have several major concerns with several experimental design caveats and the over-interpretation of the results presented here that in my opinion preclude the publication of this manuscript in *eLife* in the current form.

Major issues:

1) The effect of optogenetic silencing of APC (3s) in the DNMS task is very mild, and increases in magnitude with increasing duration of stimulation and longer delay periods between the sample and test stimuli. To my mind, the fact that inhibition of APC activity for a relatively long duration (3s) results only in a minor effect on the DNMS performance is suggesting that APC activity is actually not a major player in mediating working memory in this task in well-trained mice. Furthermore, there are several confounds in the experimental design. To assess in an interpretable fashion the effects of their manipulations, the authors should maintain the same duration of the delay (e.g. 8s), and increase the duration of optogenetic stimulation (3,6,7s, etc.). Does this change the performance of the mice? Second, to disentangle the duration of optogenetic stimulation from the natural decay of working memory (Figure 1D), it would be important to keep the same duration of optogenetic stimulation (e.g. 3s) for different durations of delay periods (6, 8, 12s) and further perform the stimulation at different times in the delay period. Third, how do we know that the first odor is removed efficiently? Do mice really need to keep its identity in memory, or is it sufficient to compare its lingering presence at low concentrations (not-detectable using a PID) to the test stimulus? While this may represent a less likely scenario, the authors should at least comment on the possibility.

2) It appears that mice have to lick in a dedicated 'response' interval in the DNMS task, and only that interval is taken as an assessment of their performance. Throughout the manuscript, it is unclear what the policy is in the case of early licks during the delay period, or the second stimulus period. In the stimulated trials, do mice lick more often in the delay period during/post stimulation? How about during the presentation of the second stimulus? The authors state that the ability of the mice to lick during the response window is not impaired, but it is important to determine whether they start to lick more outside the response window to better understand the effects of optogenetic suppression of APC. Does the optogenetic stimulation merely result in an increase in licking? (Figure 1, rate of misses decreases and of false alarms increases)? No information is provided on this.

3) Sensory perception controls (Figure 2): How long is the optogenetic stimulation? It appears to be on the shorter end of the durations used in Figure 1. Is the behavioral performance any different if optogenetic stimulation is matched accordingly to the time intervals used in Figure 1 (6s or 10s as in Figure 1 I-N)?

4) Delayed paired association (DPA) is indeed better suited for testing working memory in principle, yet the relationship between suppression APC activity and behavioral performance is weak. In Figure 4, optogenetic silencing of APC proceeds for 12s. Yet, the impairment in the performance is small (change in rate of correct choices from an Avg. ~89% to Avg. ~82%), and further appears biased by an outlier (Figure 4B). How does the observed effect change when the outlier is removed? What is the effect of decreasing the duration of optogenetic stimulation from 12s to 3s, 6s or 10s as done in the previous implementation of the DNMS task (Figure 1)? The effect of optogenetic silencing of APC on behavioral performance is even weaker or not apparent in Figure 2—figure supplement 1C, D – no change in lick efficiency, nor in sensitivity index (d'). Even in the dual task, with the GNG task embedded, the effect of optogenetic stimulation is very mild. After 9.5s of stimulation, performance in the DPA task changes from ~83% to ~78% (correct rate).

5) The neuronal correlates in APC are rather weak (Figures 5-7). There is apparent discrepancy between the behavioral performance and the neuro-correlates of activity in APC. For example, activity seems to decay steeply during the 8s delay period in terms of the fraction of neurons that are selective for the odor stimulus; Only 10% of neurons are classified as selective by the end of the 8s delay period. What is the fraction of selective neurons when the delay period is further increased (e.g. 20s)? Behavioral performance is ~80% when using the 8s delay period, and yet only a small fraction of the neurons recorded in the APC are stimulus selective (as a result the performance of the SVM classifier is substantially lower by the end of the 8s delay compared to sample stimulus).

6) Expression of the opsin does not appear to be limited to the APC, but one can observe fluorescence in several other brain areas from the slice data (Figure 1). This makes the interpretation of the results hard.

7) The learning curve for the DPA task appears to be very steep (4 days), quite similar to the GNG task. The DPA task is however rather complex, and the results reported here are unusual given experience of other labs in the field in training mice to wait for a long time (>10s) between stimuli, and furthermore embedding a second task. Could the authors include additional details on the training protocol used, and provide additional videos showing mice successfully waiting 20s or 40s in between the sample and test stimuli w/o licking impulsively?

8) Throughout the manuscript, no mechanisms are proposed on how impairing APC activity for 3s or longer does not change substantially the performance of the mice. Is the model envisioned by the authors that information is stored in some other brain areas during this interval? What is the effect of the optogenetic manipulations on the activity patterns recorded in Figures 6-8?

[Editors' note: further revisions were requested prior to acceptance, as described below.]

Thank you for resubmitting your work entitled "Active information maintenance in working memory by a sensory cortex" for further consideration at *eLife*. Your revised article has been favorably evaluated by Catherine Dulac (Senior Editor), a Reviewing Editor, and two reviewers.

The manuscript has been improved but there are some remaining issues that need to be addressed before acceptance, as outlined below.

The reviewers and I have discussed the paper. The revised version is much improved and addresses most of the earlier concerns.

1) The authors should moderate their discussions, such as the statement that they have: 'causally demonstrated that the APC is critical in the active maintenance in olfactory working memory', and that 'the APC delay activity was critical for active information maintenance in working memory with or without distraction'.

2) The authors should mention potential caveats and alternative interpretations of their data. The reviewers pointed out two such specific points, and the authors may wish to mention others.

Caveats: There could be a loop of activity involving APC and other regions. Thus suppression of activity in other parts of the brain might contribute to their observations.

Alternative interpretations: the optogenetic suppression could be partial and hence the illumination would act like a damper on recurrent activity, with a slow decay rate. This would give the observed larger effect of the longer illumination.

*Reviewer #1:*

The authors have addressed my main concerns, particularly on the spread of the illumination. The more clearly reported statistical analysis is also helpful. They have also addressed my questions about the single-trial responses.

*Reviewer #3:*

The authors thoroughly performed the new experiments and analyses suggested and nicely tightened the text. Yet, given these new results, I maintain my doubts about a strong involvement of APC in supporting working memory within the realm of the tasks (DNMS, DPA) explored in the manuscript. The reasons for my skepticism are the following:

1) Many of the behavioral effects are very small (some pass the statistical significance threshold and some don't) within the range of a few points drop in performance (<5%). For example, in Figure 2, suppressing APC for different durations at different delays from the sample stimulus has inconsistent effects. Given the analyses presented by the authors, there appears to be no significant drop in performance (Figure 2B,D) when the APC is suppressed for 3s (early, mid) or 6s (mid) during the 12s delay version of the task, or for 3s (early, mid), 6s (early, mid, late), 9s (early, mid) for the 20s delay version of the task.

2) In the fixed optogenetic suppression varied delays task, the effects of optogenetic suppression of APC are also quite variable and small: no significant change in performance for 12s and 20s conditions.

At this point, it is unclear to me whether suppression of activity in any other parts of the brain may also lead to similar size effects. In my opinion, given the data presented, it is an overstatement to assert that this work: 'causally demonstrated that the APC is critical in the active maintenance in olfactory working memory', and that 'the APC delay activity was critical for active information maintenance in working memory with or without distraction'.

---

## [Author Response]

Essential revisions:1) The authors must show the effect size of their optogenetic manipulation, as without this one cannot evaluate the results. Reviewer 2 has a detailed account of the requirements for this analysis.

We now show effect size for all optogenetic manipulation and complete statistics data, including main effects of the ANOVA, and information for non-significant ANOVA statistics. Effect size is provided in main figures (labeled above results) and figure supplement Figure 1—figure supplement 5I-M, Figure 2—figure supplement 2, Figure 3—figure supplement 1B, Figure 4—figure supplement 2 and source data for all the figures.

2) The authors must disentangle the effect of duration of stimulation vs. delay length. The reviewers ask that the authors carry out additional experiments in control and experimental protocols, using the DNMS task, in which they compare the effect of optogenetic stimulation of short 3s vs. long 12s duration, while keeping the delay the same.

We have performed the additional experiments as requested. In the first set of experiments, the delay duration of the DNMS task was fixed at 12 or 20s but the laser onset and duration were systematically changed (Figure 2A, B-C for 12s, D-E for 20s delay duration). Specifically, optogenetic suppression in the APC neuronal activity was varied in both duration (3, 6, 10, or 12s) and temporal position (early, mid or late in the delay period) in a trial-by-trial fashion. The results demonstrated that optogenetic suppression of longer laser-on duration at later phase of delay period results in larger behavioral defects. In the second set of experiments, the optogenetic suppression was kept at 3 seconds in duration and at the late delay period and the delay duration was varied between 5s and 20s, in a trial-by-trial fashion (Figure 2F). We did not observe significant interaction between delay-period duration and laser on/off (Figure 2G, Figure 2—figure supplement 1C and D, Figure 2—source data 1-2). The text has been updated accordingly.

3) The authors should address numerous points raised by the reviewers to better explain, clarify, and provide a rationale for their data. For example, they should address reviewer's concerns on whether the light is affecting other regions of the brain. This and numerous other points of clarification have been raised by the reviewers.

We made our best efforts in clarifying the issues raised by the reviewers. For example, the VGAT-ChR2 transgenic mice express ChR2 in GABAergic neurons throughout the brain. To confirm the specificity of the optogenetic suppression in APC, we have performed the *c-Fos* activity labelling experiments following optogenetic suppression. We found spatially restricted reduction in *c-Fos* labeled neurons in APC, therefore demonstrating the regional specificity in optogenetic suppression. The results have been showed in Figure 1—figure supplement 2C. We now included this, and other points of clarifications (also see following discussion) in the text.

4) Please note that eLife uses Figure Supplements (supplements to specific figures) but not a separate supplemental section. When possible, please add supplemental material to the appropriate figures, as is commonly done in eLife papers

We have now changed the formats to follow the requirements of *eLife* papers.

Reviewer #1:[…] I have a concern about how well the ChR2 optical stimulus might have been localized. The authors use an optical fiber so most of the light will indeed be directed just below the fiber end, into the APC. Nevertheless, there may be significant scatter. In Figure 1E and Figure 1— figure supplement 2 we see that expression of VGAT-ChR2 is rather broad, and there are several regions which might be affected. With this in mind, I wonder if the authors may wish to revisit their statement in the Discussion that only one out of 5 potential regions were perturbed. I worry that the core conclusion, that the sensory cortex is primarily involved in the working memory, may be weakened if several regions are affected by light scatter. One way to address this might be to obtain recordings from other relevant regions in the olfactory pathway, though I do understand that this is a substantial effort. Another possibility might be c-Fos activity labelling.

To confirm the specificity of the optogenetic suppression in APC, we have performed the *c-Fos* activity labelling experiments following optogenetic suppression. We found spatially restricted reduction in *c-Fos* labeled neurons in APC, therefore demonstrating the regional specificity in optogenetic suppression. The results have been showed in Figure 1—figure supplement 2C.

The authors also perform single unit recordings and demonstrate sustained activity through the delay period in about 10% of neurons. While this seems reasonably clear, it would be nice to have a more complete analysis here. How many do this on single-trials? Is there more we might understand from a single-trial analysis? What might we infer from a closer comparison of 4-s vs. 8-s delay recordings?

We have included more detailed single-trial analysis based on auROC analysis, which is now showed in the updated Figure 5C and D, and Figure 5—figure supplement 2A-K. We also performed a closer comparison of 4s- and 8s-delay recordings, and the results are showed in Figure 5L and Figure 5—figure supplement 3F and G.

Reviewer #2:This work addresses a critical limitation of so-called sensory recruitment model – namely that there is no causal evidence that sensory delay activity is involved in WM. This paper is timely, the experiments are thorough and address nearly all potential objections, and the results are clear. Overall, this is an excellent paper. However, there are a few concerns that the authors need to address that could potentially affect the interpretation of the results. Beyond that, the clarity in presentation of the results, particularly when it comes to the organization of the figures, is lacking and needs quite a bit of work.1) The authors do not include any measures of effect size, indicate the main effects of the ANOVA, or provide sufficient information for non-significant ANOVAs. These all hinder the ability to evaluate the results. For example, the missing effect sizes don't allow for a proper comparison between the main analyses and the control analyses (e.g., Figure 1 vs. Figure 2). Additionally, it seems possible that the light on vs. off had an effect on control mice as well (albeit a smaller one). If this were true, it would require further discussion.

We now showed effect size for all optogenetic manipulation and complete statistics data, including main effects of the ANOVA, and information for non-significant ANOVA statistics. Effect size is provided in main figures (labeled above results) and Figure 1—figure supplement 5I-M, Figure 2—figure supplement 2, Figure 3—figure supplement 1B, Figure 4—figure supplement 2 and source data for all the figures.

In longer delay tasks with long optogenetic stimulation (e.g. 12s delay with 10s optogenetic stimulation), we indeed observed small but significant effect of light on vs. off trials, which may be due to distraction induced by light itself (performance correct rate for 12s delay duration). Such effect can be controlled by the interaction statistics between genotype and laser on/off, which was commonly observed. In the newly added experiments (Figure 2), a blue light mask was applied to diminish the visual distraction. Statistics are included in the Figure 1J and discussion about this was included in the text (subsection “APC delay-period activity is critical in olfactory DNMS”).

2) The authors need to address one possible alternate interpretation of their results. Could it be the case that APC perturbation disrupted the delay activity of downstream neurons (e.g., in PFC) that are critical for the memory? How are these results inconsistent with that view?

It is likely that the delay activity of other downstream brain regions of sensory regions may contribute to the maintenance of sensory information in a working memory task. The existence of these regions does not conflict with the current results that a sensory region plays important roles in WM maintenance, because this hypothetical downstream brain region critically depends on the APC delay activity to play its role. The exact mechanism underlying the distributed neural circuit underlying WM remains to be determined in the future. We do not believe that mPFC is important for well-trained mice, because our previous work has demonstrated that mPFC delay activity are not critical in the DNMS task once the mice were well-trained (Liu et al., 2014). We have added more discussion to this point (e.g., subsection “Distributed network Interaction”).

3) The authors' claim rests on the fact that light suppressed the activity of APC neurons in the VGAT-ChR2 mice. However, as shown in Figure 1F and Figure 1—figure supplement 4, there are a variety of responses to the light stimulus. In order to support their claims, the authors need to address this issue directly. For example, a statement clarifying that a wide majority of neurons demonstrated optogenetic suppression, or a simple analysis demonstrating that more neurons were suppressed that would be expected by chance would help advance the claim that the net effect of the manipulation was suppression.

We have now clarified that a wide majority of neurons (79%) demonstrated optogenetic suppression. This information is also updated in Figure 1.

Reviewer #3:[…] 1) The effect of optogenetic silencing of APC (3s) in the DNMS task is very mild, and increases in magnitude with increasing duration of stimulation and longer delay periods between the sample and test stimuli. To my mind, the fact that inhibition of APC activity for a relatively long duration (3s) results only in a minor effect on the DNMS performance is suggesting that APC activity is actually not a major player in mediating working memory in this task in well-trained mice. Furthermore, there are several confounds in the experimental design. To assess in an interpretable fashion the effects of their manipulations, the authors should maintain the same duration of the delay (e.g. 8s), and increase the duration of optogenetic stimulation (3,6,7s, etc.). Does this change the performance of the mice? Second, to disentangle the duration of optogenetic stimulation from the natural decay of working memory (Figure 1D), it would be important to keep the same duration of optogenetic stimulation (e.g. 3s) for different durations of delay periods (6, 8, 12s) and further perform the stimulation at different times in the delay period. Third, how do we know that the first odor is removed efficiently? Do mice really need to keep its identity in memory, or is it sufficient to compare its lingering presence at low concentrations (not-detectable using a PID) to the test stimulus? While this may represent a less likely scenario, the authors should at least comment on the possibility.

1) The 3s optogenetic silencing effect of APC is indeed smaller comparing to 6s and 1s optogenetic silencing effects. One possibility is due to the regionally specific optogenetic silencing (small area with reduction in *c-Fos* staining, Figure 1—figure supplement 2C). Because our optogenetic suppression is focal, APC areas not affected by laser might still maintain the information. Moreover, other studies have revealed dependence of optogenetic effect on delay duration. For example, Gorden lab’s study (Bolkan et al., 2017) showed that optogenetic silencing mPFC impaired spatial WM performance in task with 60 s but not 10 s delay duration.

2) We have performed the additional experiments as requested. In the first set of experiments, the delay duration of the DNMS task was fixed at 12 or 20s but the laser onset and duration were systematically changed (Figure 2A, B-C for 12s, d-e for 20s delay duration). Specifically, optogenetic suppression in the APC neuronal activity was varied in both duration (3, 6, 10, or 12s) and temporal position (early, mid or late in the delay period) in a trial-by-trial fashion. The results demonstrated that optogenetic suppression of longer laser-on duration at later phase of delay period results in larger behavioral defects. In the second set of experiments, the optogenetic suppression was kept at 3 seconds in duration and at the late delay period and the delay duration was varied between 5s and 20s, in a trial-by-trial fashion (Figure 2F). We did not observe significant interaction between delay-period duration and laser on/off (Figure 2G, Figure 2—figure supplement 1C and D, Figure 2—source data 1-2). The text has been updated accordingly.

3) As the reviewer pointed out, mice are not likely to use the low concentration residual odor to perform the task. We have previously showed the sensory threshold of mice in a working memory task in the experiments setup (Liu et al., 2014), which is higher than the residual concentration during most of the delay period. Furthermore, if the mice depend on residuals to perform the DNMS task, the dual-task performance would be much worse than the experiment data since any lingering residual would be flushed by the many-orders more concentrated olfactory cue for the inner task. We have included more discussion in the text (subsections “The design and performance of a delayed non-match to sample task” and “APC delay activity is critical for active maintenance against a distracting task”).

2) It appears that mice have to lick in a dedicated 'response' interval in the DNMS task, and only that interval is taken as an assessment of their performance. Throughout the manuscript, it is unclear what the policy is in the case of early licks during the delay period, or the second stimulus period. In the stimulated trials, do mice lick more often in the delay period during/post stimulation? How about during the presentation of the second stimulus? The authors state that the ability of the mice to lick during the response window is not impaired, but it is important to determine whether they start to lick more outside the response window to better understand the effects of optogenetic suppression of APC. Does the optogenetic stimulation merely result in an increase in licking? (Figure 1, rate of misses decreases and of false alarms increases)? No information is provided on this.

Now we provide the lick rate throughout the entire trials of different optogenetic design in the updated Figure 1—figure supplement 6A-E. As show in the figures the optogenetic suppression did not merely result in a lick increase in response period, especially when the timing of optogenetic suppression was varied (Figure 1—figure supplement 6A and B). A small increase in lick rate was observed immediately after the laser offset in the fixed optogenetic suppression, likely due to the association between stimulation-offset and test-onset. The differences are small and transient, which was unlikely to affect the quantification of behavioral response during the response window. (Figure 1—figure supplement 6, C, D and E). We have now included the results in the figure and discussed in the text (subsection “APC delay-period activity is critical in olfactory DNMS”).

3) Sensory perception controls (Figure 2): How long is the optogenetic stimulation? It appears to be on the shorter end of the durations used in Figure 1. Is the behavioral performance any different if optogenetic stimulation is matched accordingly to the time intervals used in Figure 1 (6s or 10s as in Figure 1 I-N)?

We performed extra experiments to better control for sensory perception. In the previous manuscript, the DNMS baseline optogenetic stimulation duration is 3s which matched the delay stimulation in the 5s-delay DNMS task. We added similar control experiments with 6s and 10s optogenetic stimulation durations that matched the delay stimulations in the 8s- and 12s-delay DNMS task. The performance was not impaired in these tasks (updated Figure 3C).

4) Delayed paired association (DPA) is indeed better suited for testing working memory in principle, yet the relationship between suppression APC activity and behavioral performance is weak. In Figure 4, optogenetic silencing of APC proceeds for 12s. Yet, the impairment in the performance is small (change in rate of correct choices from an Avg. ~89% to Avg. ~82%), and further appears biased by an outlier (Figure 4B). How does the observed effect change when the outlier is removed? What is the effect of decreasing the duration of optogenetic stimulation from 12s to 3s, 6s or 10s as done in the previous implementation of the DNMS task (Figure 1)? The effect of optogenetic silencing of APC on behavioral performance is even weaker or not apparent in Figure 2—figure supplement 1 C, D – no change in lick efficiency, nor in sensitivity index (d'). Even in the dual task, with the GNG task embedded, the effect of optogenetic stimulation is very mild. After 9.5s of stimulation, performance in the DPA task changes from ~83% to ~78% (correct rate).

The optogenetic silencing impairment in DPA performance is smaller than that in DNMS task (comparing Figure 4 with Figure 1). The task designs between two tasks are quite different, for example DPA task require the association between sample and test odors, which is not required for DNMS task. Therefore, the neural circuit for the DPA task might be different from that for the DNMS task, which remains to be determined in future studies. We have added discussion in the text (subsection “Debate about the necessity of delay-period activity in sensory cortex for WM”).

We believe that the extra two experiments we already performed in Figure 2A-E already demonstrated the significant interaction between delay duration and laser-on duration. Although the similar experiments for the DPA task do provide more information, these will further delay the revision for about two more months. We leave the reviewers and reviewing editor to decide whether it is absolutely necessary.

We find no outliers in the DPA optogenetic impairment in performance (difference in correct rate between laser-on and laser-off trials), with two widely used statistic methods: 1) for elements more than 1.5 interquartile ranges above the upper quartile or below the lower quartile(Hattori et al., 2017; Kato, Gillet, Peters, Isaacson and Komiyama, 2013; Sreenivasan et al., 2016); 2) the Grubb’s method (Burgos-Robles et al., 2017; Carus-Cadavieco et al., 2017; Namburi et al., 2015). We have now included the results in the text.

5) The neuronal correlates in APC are rather weak (Figures 5-7). There is apparent discrepancy between the behavioral performance and the neuro-correlates of activity in APC. For example, activity seems to decay steeply during the 8s delay period in terms of the fraction of neurons that are selective for the odor stimulus; Only 10% of neurons are classified as selective by the end of the 8s delay period. What is the fraction of selective neurons when the delay period is further increased (e.g. 20s)? Behavioral performance is ~80% when using the 8s delay period, and yet only a small fraction of the neurons recorded in the APC are stimulus selective (as a result the performance of the SVM classifier is substantially lower by the end of the 8s delay compared to sample stimulus).

We observed only about 10% APC neurons carried WM information during the late delay period (Figures 5G, 5I). The smaller percentage of APC neurons in coding the maintained information was consistent with the sparse coding previously observed in piriform cortex37,38,40 (but see 45). Because 10% APC neurons still correspond to quite large number neurons (APC was estimated to be composed of 500,000 neurons, Stevens et al., 2017), we do not believe the 10% selective neurons contradict with the 80% correct rate in performance. We have added discussion of this point in the text (Discussion section).

6) Expression of the opsin does not appear to be limited to the APC, but one can observe fluorescence in several other brain areas from the slice data (Figure 1). This makes the interpretation of the results hard.

To confirm the specificity of the optogenetic suppression in APC, we have performed the *c-Fos* activity labelling experiments following optogenetic suppression. We found spatially restricted reduction in *c-Fos* labeled neurons in APC, therefore demonstrating the regional specificity in optogenetic suppression. The results have been showed in Figure 1—figure supplement 2C.

7) The learning curve for the DPA task appears to be very steep (4 days), quite similar to the GNG task. The DPA task is however rather complex, and the results reported here are unusual given experience of other labs in the field in training mice to wait for a long time (>10s) between stimuli, and furthermore embedding a second task. Could the authors include additional details on the training protocol used, and provide additional videos showing mice successfully waiting 20s or 40s in between the sample and test stimuli w/o licking impulsively?

The GNG only needs 100 trials within one day of training to learn, as already shown in Figure 3—figure supplement 1, red curve. We have updated the text to clarify this point. Our automatic training system, as well as the training protocol design, have been specifically optimized to facilitate the rapid learning of working memory tasks. Additional details on the training protocol was included in the updated text, and more complete details are available in a previous publication (Han et al., 2019). We also plotted the licking during a longer-delay (20s) DNMS task (Figure 1—figure supplement 6A), and included a 5 min continuous video clip of mice performing the DNMS task with 20s delay.

8) Throughout the manuscript, no mechanisms are proposed on how impairing APC activity for 3s or longer does not change substantially the performance of the mice. Is the model envisioned by the authors that information is stored in some other brain areas during this interval? What is the effect of the optogenetic manipulations on the activity patterns recorded in Figures 6-8?

The optogenetic stimulation impairment is mild yet significant in shorter-delay DNMS tasks and in the DPA design tasks. This is probably due to the incomplete coverage of optogenetic stimulation of the APC in the mice, since the APC is very large in size. It is also possible that other brain regions beyond APC could contribute to the information maintenance in the tasks. We added more discussion of this point in the text (subsection “Debate about the necessity of delay-period activity in sensory cortex for WM”).

In the current design, the optrodes maximized the recording of nearby optogenetic-stimulation manipulated neurons (Figure 1F, 90%, 94 out of 104), thus the recorded neuron activity would be completely disrupted as long as the optogenetic stimulation is present. Recording neuron activity in other part of the APC and other brain regions while one part of the APC is optogenetically suppressed lies beyond the scope of the current study. We will address this point in future.

[Editors' note: further revisions were requested prior to acceptance, as described below.]

The manuscript has been improved but there are some remaining issues that need to be addressed before acceptance, as outlined below.The reviewers and I have discussed the paper. The revised version is much improved and addresses most of the earlier concerns.1) The authors should moderate their discussions, such as the statement that they have: 'causally demonstrated that the APC is critical in the active maintenance in olfactory working memory', and that 'the APC delay activity was critical for active information maintenance in working memory with or without distraction'.2) The authors should mention potential caveats and alternative interpretations of their data. The reviewers pointed out two such specific points, and the authors may wish to mention others.Caveats: There could be a loop of activity involving APC and other regions. Thus suppression of activity in other parts of the brain might contribute to their observations.Alternative interpretations: the optogenetic suppression could be partial and hence the illumination would act like a damper on recurrent activity, with a slow decay rate. This would give the observed larger effect of the longer illumination.

We have moderated the discussions according to your and the reviewers’ suggestions and removed the words such as “causally” and “critically”. We also described the potential caveats and alternative interpretations of our data suggested by the reviewers, such as small effect size and small laser illumination area. In particular, we specifically added a session in discussion about “Effect size of optogenetic suppression” and modified the subtitles of results. Furthermore, we modified the text for other caveats, for example we added ‘partially’ in describing function transfer from mPFC to APC during learning.